



# Effects of eutrophication on sedimentary organic carbon cycling in five temperate lakes

Annika Fiskal[1*], Longhui Deng[1], Anja Michel[1], Philip Eickenbusch[1], Xingguo Han[1], Lorenzo Lagostina[1], Rong Zhu[1], Michael Sander[1], Martin H. Schroth[1], Stefano M. Bernasconi[3], Nathalie Dubois[2,3], Mark A. Lever[1*]

[1] Institute of Biogeochemistry and Pollutant Dynamics (IBP), *ETH Zurich, Universitätstrasse 16, 8092 Zurich, Switzerland*
[2] Surface Waters Research – Management, Eawag, Swiss Federal Institute of Aquatic Science and Technology, Überlandstrasse 133, 8600 Dübendorf, Switzerland
[3] *Department of Earth Sciences, ETH Zurich, Sonneggstrasse 5, 8092 Zurich, Switzerland*

[*]*Correspondence to*: Annika Fiskal (annika.fiskal@usys.ethz.ch) and Mark A. Lever (mark.lever@usys.ethz.ch)

**Keywords**
*organic carbon burial, organic carbon accumulation, temperate lake sediment, microbial respiration, phosphorus, methanogenesis, redox zonation, power model, oxygen exposure time*

**Abstract.** Even though human induced eutrophication has severely impacted temperate lake ecosystems over the last centuries, the effects on total organic carbon (TOC) burial and mineralization are not well understood. We study these effects based on sedimentary records from the last 180 years in five Swiss lakes that differ in trophic state. We compare changes in content of TOC and modeled TOC accumulation rates through time to historical data on algae blooms, water column anoxia, wastewater treatment, artificial lake ventilation, and water column phosphorus (P) concentrations. We furthermore investigate the effects of eutrophication on rates of microbial TOC remineralization and vertical distributions of microbial respiration reactions in sediments. Our results indicate that the history of eutrophication is well reflected in the sedimentary record. Subsurface peaks in sedimentary TOC coincide with past periods of elevated P concentrations in lake water. Sediments of eutrophic lakes show overall higher rates of microbial respiration, and a higher relative contribution of methanogenesis to total respiration. Yet, a clear impact of lake trophic state on the zonation of microbial respiration reactions is absent. Moreover, even though water column P concentrations have been reduced by ~80% (range: ~50-90%) since the period of peak eutrophication in the 1970s, TOC burial and accumulation rates have only decreased significantly (~20 and 25%) in two of the five lakes. Hereby we found no clear relationship between the magnitude of the decrease in P concentrations and the change in TOC burial and accumulation rate. Instead, artificial lake ventilation, which is used to prevent water column anoxia in eutrophic lakes, may help sustain high rates of TOC burial and accumulation in sediments despite strongly reduced water column P concentrations. Our results provide novel insights into how eutrophication and eutrophication management practices affect organic carbon burial and the distribution of microbial respiration reactions in temperate lakes. These insights are important to understanding how anthropogenic activities affect the size of the carbon pool that is stored globally in lacustrine sediments.



# 1 Introduction

Lake sediments play an important role in the global carbon cycle despite covering only about 1 % of the Earth's surface area (Cole et al., 2007; Battin et al., 2009). Estimates of global carbon dioxide ($CO_2$) net fluxes from lake systems range from 60 to 580 Tg C $yr^{-1}$ (Raymond et al., 2013; Holgerson and Raymond, 2016). Methane ($CH_4$) fluxes are much smaller (6 to 36 Tg C;

(Bastviken et al., 2004), but are also globally important given that $CH_4$ has a 28 to 36 times higher global warming potential than $CO_2$ on a timescale of about 100 years (Cubasch et al., 2014). Besides being important sources of $CO_2$ and $CH_4$, lakes are also important sinks of organic C (OC). Global storage of C in lake sediments for the entire Holocene is estimated to be ~820 Pg C (Einsele et al., 2001; Mendonca et al., 2017)

       Whether OC in lake sediments is buried and preserved over geologic time or degraded to $CO_2$ and $CH_4$ is partially

controlled by sediment microorganisms, which break down OC for energy conservation and biosynthesis (Nedwell, 1984). In a first step, microorganisms enzymatically hydrolyze organic macromolecules, e.g. proteins, nucleic acids, lipids and polysaccharides, extracellularly into sufficiently small components that can be taken up across the cell membrane (Canfield et al., 2005). If oxygen ($O_2$) is available as an electron acceptor, organisms performing initial hydrolysis are often capable of oxidizing hydrolysis products all the way to $CO_2$. In the absence of $O_2$, OC is degraded to $CO_2$ or $CH_4$ by multiple groups of

more specialized microorganisms (Canfield et al., 2005). Primary fermenters carry out the initial extracellular hydrolysis, and subsequently gain energy from the intracellular disproportionation of hydrolysis products to smaller molecules such as $H_2$, short chain organic acids, and alcohols. In some cases, an intermediary secondary fermentation step takes place whereby the organic acids and alcohols produced by primary fermenters are disproportionated to $H_2$, acetate, and $C_1$ compounds (Capone and Kiene, 1988; Schink, 1997). The products of primary and secondary fermentation are subsequently converted to $CO_2$ or

$CH_4$ by respiring organisms using - in order of energy yields from high to low - nitrate ($NO_3^-$), manganese(IV) (Mn(IV)), ferric iron (Fe(III)), sulfate ($SO_4^{2-}$), and $CO_2$ as electron acceptors (Froelich et al., 1979; Drake et al., 2006). The differences in energy yields can result in a vertical zonation ("redox zonation") of respiration reactions, with $NO_3^-$ reduction (denitrification) occurring near the sediment surface, as soon as $O_2$ is depleted, $CO_2$ reduction via methanogenesis dominating in deeper layers, where all other electron acceptors have been depleted, and Mn(IV), Fe(III), and $SO_4^{2-}$ reduction taking place

in distinct sediment intervals between the zones of denitrification and methanogenesis (Canfield et al., 2005; Canfield and Thamdrup, 2009).

       A complex set of interacting variables determines whether microorganisms break down sedimentary organic compounds, and thus whether organic carbon is stored in sediment in the long term, or returns to the hydrosphere and atmosphere as $CO_2$ and methane over shorter time scales. Aerobic respiration results in the breakdown of a larger fraction of

the sedimentary organic carbon pool compared to anaerobic processes (Lehmann et al., 2002; Sobek et al., 2009; Katsev and Crowe, 2015). The $O_2$ exposure time, i.e. the time each organic molecule is present under oxic conditions, is also positively correlated with organic matter degradation. This $O_2$ exposure time is affected by the presence of sediment macrofauna, which physically transport sediment from deeper anoxic layers to the surface or pump $O_2$ into deeper anoxic layers (Kristensen, 2000;





Meysman et al., 2006; Kristensen et al., 2012; Middelburg, 2018). Macrofauna also mineralize OC through their own feeding activities, and stimulate the microbial remineralization of organic matter by breaking large organic particulates into smaller ones with a larger surface area for microbial colonization. The chemical composition and structure of the OC also affects OC preservation. OC of terrestrial plants is typically more resistant to microbial attack than OC of phytoplankton due to presence of biologically resistant polymers, e.g. lignin, waxes, resins, that are absent in phytoplankton (Burdige, 2007). Abiotic processes, e.g. condensation reactions (e.g. Burdige, 2007), incorporation of sedimentary Fe(III) (Lalonde et al., 2012) or sulfur compounds (Werne et al., 2000; Hebting et al., 2006; Werne et al., 2008), and physical protection by adsorption or encapsulation can also significantly decrease rates of microbial OC degradation (reviewed in (Hedges et al., 2000)).

Besides these natural factors, the burial of C in lake sediment is also largely influenced by anthropogenic activities, such as agriculture, wastewater input, and urbanization (Heathcote and Downing, 2012; Anderson et al., 2013). OC burial rates may increase due to increased water column primary production resulting from increased nutrient input, especially P (Dean and Gorham, 1998; Maerki et al., 2009; Heathcote and Downing, 2012; Anderson et al., 2013; Anderson et al., 2014), which is the limiting nutrient for primary production in most lake water columns (Correll, 1999; Conley et al., 2009). Hereby increased OC loading, due to stimulation of primary production, enhances OC sedimentation rates and microbial $O_2$ consumption (Valiela et al., 1997; Gontikaki et al., 2012). The resulting (seasonal) water column hypoxia or anoxia negatively impact ecosystem functioning, as well as commercial and recreational uses (Valiela et al., 1997; McGlathery, 2001; Breitburg, 2002). Strategies to manage eutrophication include restrictions in terrestrial P use, agriculture-free buffer zones, wastewater treatment with P precipitation systems, and artificial water column mixing and aeration (Schindler, 2006; Conley et al., 2009). These remediation strategies can lead to substantial reductions in P concentrations in lake water (Müller et al., 1990; Liechti, 1994). Yet, there are still open questions regarding the influence of these mitigation strategies on OC burial, in part due to P retention and release from lake sediments (Gachter and Muller, 2003; Moosmann et al., 2006). Furthermore, the effects of lake-specific variables, e.g. lake area, water depth, water residence time, sedimentation rates, and sediment focusing (Lehman, 1975; Blais and Kalff, 1995), on the success of eutrophication mitigation practices remain unclear.

Here we investigate organic C burial through time in five temperate lakes that have, to variable degrees, been subjected to anthropogenic eutrophication over the past 180 years. We hypothesize that *(i)* OC burial rates have historically increased in response to anthropogenic eutrophication, and that *(ii)* increased P concentrations, and its effects on water column primary production and $O_2$ concentrations, are the main driver of OC burial. We furthermore hypothesize that *(iii)* microbial respiration rates are higher in eutrophic lake sediment due to higher OC availability, and that *(iv)* due to changes in microbial respiration rates, the distribution of dominant respiration processes shows a clear response to changes in trophic state. Our results indicate that anthropogenic eutrophication has indeed increased OC burial and increased total OC respiration rates, especially methanogenesis rates, and cell-specific microbial C turnover. Yet, we observe no clear impact of eutrophication on the zonation of microbial respiration reactions relative to each other, or on microbial population size. Furthermore, despite strong decreases in water column P concentrations to pre-eutrophication levels in recent years, TOC burial rates remain clearly elevated.



## 2 Material and Methods

### 2.1 Study sites & their eutrophication history

We sampled sediments from three water depths in five lakes in central Switzerland: Lake Lucerne, Lake Zurich, Lake Zug, Lake Baldegg, and Lake Greifen (Fig. S1). All 5 lakes undergo strong seasonal changes in primary production, thermal

stratification, and sedimentation (Teranes et al., 1999a; Teranes et al., 1999b; Buergi and Stadelmann, 2000; Naeher et al., 2013), but differ in land use and in histories of anthropogenic eutrophication and eutrophication mitigation (Fig.1, for further info see Fig. S2, SI Text 1 "Extended Lake description"). Based on total P concentrations in surface water today, Lake Lucerne is oligotrophic, Lake Zurich is mesotrophic, and the other three lakes are all highly eutrophic.

Over the past 180 years, the five lakes show similar, but distinct trends in eutrophication history (Fig. 1). In Lake

Baldegg, increases in wastewater input, industrial use, and agriculture led to strong P concentration increases, algal blooms and water column anoxia already around 1890. Such changes did not occur in Lake Greifen until the 1920s. In Lake Zug a slower but steady increase in P concentrations started in the 1930s, and coincided with algal bloom occurrence. For Lake Zurich no reconstructed P concentrations are available, however, measured concentrations indicate relatively low and steady P levels from 1950 to 1960, followed by an increase from thereon. In Lake Lucerne reconstructed P concentrations indicate

low and relatively steady concentrations until a small, temporary increase started in the 1960s.

In all five lakes, the strongest increases in P concentrations occurred between ~1950–1970, yet, the magnitude of these increases differed greatly (Fig. 1). Annual P concentration maxima in Lake Greifen (16.1 µM, 1969) and Lake Baldegg (15.4 µM, 1969) were ~2.5 fold higher than in Lake Zug (6.6 µM, 1980), ~5 fold higher than in Lake Zurich (3.3 µM, 1973), and ~9 fold higher than in Lake Lucerne (1.7 µM, 1973). Since then, remediation measures, such as wastewater treatment, and

P detergent bans have strongly decreased water column P concentrations. Today the highest P concentrations are found in Lake Zug, followed by Lake Baldegg, Lake Greifen, Lake Zurich, and Lake Lucerne.

### 2.2 Sampling

All took place in June and July of 2016. Three sublittoral stations differing in water depth and bottom water $O_2$ concentrations were sampled (Table 1). All sites were bioturbated, based on presence of infaunal chironomid larvae and/or oligochaete worms,

except the deepest station in the hypoxic basin of Lake Zurich (137 m water depth).

Per each lake station, one 60-mm diameter and 3–4 150-mm diameter gravity cores (UWITEC, AT) were taken from boats or motorized platforms. The thin cores were used for analyses of radionuclides, X-ray fluorescence, grain size, and archiving (one-half) (for core photos see SI Fig. S3). Wide cores underwent a brief lithostratigraphic description and were then used as follows: the core with the least disturbed sediment surface was used for microsensor measurements ($O_2$, pH). Sediment

porewater samples were obtained by rhizons (0.2 µm pore size, Rhizosphere) from a designated core with pre-drilled holes that were taped during coring. The initial dead spaces of syringes and stop cocks were flushed twice with the first 2–3 mL of porewater to remove any air ($O_2$). The porewater was then sampled under strictly anoxic conditions to enable downstream



analyses on dissolved anions and cations including redox sensitive elements, such as dissolved iron ($Fe^{2+}$) and hydrogen sulfide ($HS^-$). Sediment samples for cell counts, methane concentration analyses, and physical property determinations (porosity, bulk density, dry density, water content) were taken from a third core using sterile cut-off syringes. Samples for solid-phase carbon analyses (TOC, total inorganic carbon (TIC)) were also taken from this core. Macrofauna was collected by sieving a fourth

core or the core previously used for microsensor analyses through 400 and 200 µm meshed sieves, and preserved in 70 % ethanol.

### 2.3 Analyses

***Microsensor analyses.*** To determine the distribution of aerobic microbial activity and pH conditions in surface sediments, depth profiles of dissolved [$O_2$] and pH were measured with 100-µm Clark-type microelectrodes using a field multimeter

system and micromanipulator (Unisense, DK). Cores were fixed in tall plastic boxes that had been filled with lake water and cooled to bottom water temperatures, which had been measured immediately after core retrieval. Overlying water was bubbled with air throughout the measurements to prevent establishment of anoxic conditions. Three profiles were measured at different locations in each core at a minimum vertical resolution of 100 µm.

***[$Fe^{2+}$] and [$Mn^{2+}$].*** Porewater concentration profiles of $Fe^{2+}$ and $Mn^{2+}$ were measured to determine the distributions of microbial Fe and Mn reduction. 3–5 ml of porewater were fixed with 50 µl of 30 % HCl and kept at 4°C until measurement by Inductively Coupled Plasma-Optical Emission Spectroscopy (ICP-OES) (5100, Agilent Technologies) after dilution with Milli-Q water. Standards were made using ICP-multi element standards (solution IV, MERCK, Certipur). Dissolved $Fe^{2+}$ was also determined spectrophotometrically (Plate Reader Biotek, Synergy HT) at 562 nm absorption using a ferrozine assay

(Stookey, 1970; Braunschweig et al., 2012). Standards were made from $FeCl_2$ (Sigma, analytical quality) and verified using ICP-OES. Both methods show good general agreement (Fig. S4), however, values obtained by spectrophotometry were often slightly lower than those measured by ICP-OES. Tests involving different ferrozine concentrations, spiked samples, and various centrifugation treatments showed that ICP-OES data were less prone to matrix effects and overall more robust. We thus only discuss the $Fe^{2+}$ data measured by ICP-OES from here on.

***[$HS^-$].*** Porewater concentrations of HS- were measured to detect sulfate-reducing microbial activity and potential coupling between microbial sulfate and metal reduction. 1 ml of porewater was fixed with 1 ml of 5 % zinc acetate and frozen at -20°C until photometric determination on a plate reader (Synergy HT, BioTek) with diamine reagent (methylene blue method (Cline, 1969)). Standards were prepared from a zinc sulfide stock solution made from sodium sulfide diluted in 1 % zinc acetate

solution under precipitation of zinc sulfide. $HS^-$ concentrations were then determined by titration with 25 mM sodium thiosulfate.





***Inorganic ions.*** Concentrations of the inorganic anions (sulfate ($SO_4^{2-}$) and nitrate ($NO_3^-$), and the cations ammonium ($NH_4^+$) were quantified to determine the distribution of microbial sulfate reduction and N cycling (denitrification, breakdown of N-containing organic matter). For each analysis 3 ml of porewater were sampled. 5 µl of NaOH (2M) and 5 µl HCl (2M) were added for anion and cation analyses, respectively, to increase or lower the pH by 2–3 units. Samples were kept on ice in the field and frozen at -20 °C until measurement by Ion chromatography (DIONEX DX-320 for Anions, DX-ICS-1000 for Cations.). Standards were prepared by dissolving analytical quality sodium nitrate, sodium sulfate, and ammonium chloride in Milli-Q water. The DX-320 was equipped with a AERS 500, a 4-mm suppressor, a guard column (AG11-HC), and a main column (AS11-HC). The eluent, potassium hydroxide (10 mM), was generated by an eluent generator cartridge. The DX-ICS-1000 was equipped with an SCRS Ultra II, a 4-mm suppressor, a CG 12 A guard column, and a CS 12 A main column. The eluent used was methanosulfonic acid (20 mM) and prepared fresh on every measurement day.

***Dissolved inorganic carbon (DIC)***. DIC concentration profiles were measured to determine distribution and rates of OC mineralization. Porewater was filled headspace-free into 1.5 ml borosilicate vials and stored at 4°C for up to 4 weeks. Samples were analyzed along with standards, produced by dissolving sodium bicarbonate in Milli-Q, on a GasBench II (Thermo Fisher Delat V and TC/EA) coupled to MS after acidification with 85 % phosphoric acid.

***[CH4].*** Concentration profiles of $CH_4$ were measured to determine the distribution and rates of microbial methanogenesis. 2 $cm^3$ of sediment were transferred to 20-ml crimp vials containing 2.51 g NaCl and 5 ml Milli-Q water, crimped, thoroughly homogenized by shaking, and stored on ice until analysis by Gas Chromatography (GC). The GC was equipped with an autosampler (PAL GC-xt, CTC Analytics AG) and a Flame Ionization Detector (FID) (Thermo Scientific™ TRACE™ Ultra Gas Chromatograph). The working conditions for the measurement were: prep flow 10 pounds per square inch (psi), GC flow 14 psi, and $CO_2$ reference gas 25 psi.

***Solid-phase iron pools.*** For extraction of biologically available particulate Fe(II) and Fe(III), 2 $cm^3$ of sediment were transferred to 20-ml crimp vials, immediately flushed with $N_2$, and stored cool at 4°C until measurement. Extraction from 0.2–0.6 g of wet sediment was performed with 5 ml 0.5 M HCl. The extract was split into two subsamples. To determine Fe(II), 40 µl of extract were mixed with 2 ml of 0.02 % ferrozine in 50 mM HEPES at pH 7 (Stookey,1970 and Lovley and Phillips 1987) and then quantified photometrically on a Plate Reader (Synergy HT, BioTek). For total Fe, 1 ml of extract was mixed with 0.2 ml hydroxylamine (1.5 M) to reduce Fe(III) to Fe(II) and then measured as above. Fe(III) concentrations were the difference of total Fe and Fe(II). Standards consisted of dilution series of 100 ml 100 mM $FeSO_4$ in 0.5 M HCl.

***Redox state of sediment.*** Total electron accepting and donating capacities of the sediment were determined by mediated electrochemical reduction and oxidation, respectively (Kluepfel et al., 2014; Klupfel et al., 2014; Sander et al., 2015). Samples were treated as for solid-phase Fe extraction and analyzed inside an anoxic glove box. Before measurement, each sample was





diluted into a slurry by adding 10 ml of $O_2$-free Milli-Q water. Further details can be found in the SI Text "Detailed description of redox state analysis of the sediment".

*TOC.* TOC was measured to determine OC accumulation and burial over time. 5–10 g of frozen sediment was freeze-dried in glass vials, homogenized, and split into three subsamples. One subsample was decarbonized with 6 M HCl, oven-dried, and
homogenized for TOC analyses with an elemental analyzer (Thermo Fisher Flash EA 1112) coupled to an isotope-ratio-mass spectrometer (Thermo Fisher Delta V Plus) (EA-IRMS). The second subsample was analyzed directly by EA-IRMS to determine Total Carbon (TC) content in order to calculate TOC as the difference of TC minus TIC. Standards consisted of known amounts of peptone, atropine, and nicotinamide. The third subsample was used to determine TIC by coulometry (5011 $CO_2$ Coulometer). Some TC measurements could not be used due to high sulfur contents in the samples; therefore, TOC was
back-calculated from TOC % of the decarbonized fraction using an acidification factor derived from TIC measurements for all samples. To check that these calculations were correct, TOC was also calculated as the difference between TC minus TIC for those samples where TC measurements were available.

*Cell counts.* Cell counts were performed on 0.5 $cm^3$ of sediment after cell extraction by combined sonication and fluoric acid treatment followed by flow cytometric quantification after the method of Deng et al. (2019).

*Physical Properties Analyses.* Porosity, water content, bulk density, and dry density of sediment samples were determined by weighing 2 $cm^3$ of wet sample before and after oven drying at 60°C for at least 48 h.

*Determination of sedimentation rates.* Analyses of unsupported $^{210}Pb$ (half-life: 22.3 years) and $^{137}Cs$ (30.2 years) were performed by gamma spectroscopy on 2–20 g of dry sediment subsamples on a High-purity Germanium well detector. Calibrations were performed using the National Institute of Standards and Technology NG7 standard (246 kBq). Raw data
were processed using the software GENIE2K 3.2.1. Sedimentation rates based on $^{210}Pb$ were calculated according to (Binford, 1990) Sedimentation rates $^{137}Cs$ ($cm\ yr^{-1}$) were calculated using the distinct peaks from bomb testings (1954, 1963) and the Chernobyl accident (1986). Surface sediments were assigned the year 2016, as confirmed by the short-lived radionuclide Beryllium-7 (half-life: 53.3 days). Sedimentation rate estimates based on $^{137}Cs$ and $^{210}Pb$ were similar throughout the dataset (the coefficient of variation for all stations was <10%, the mean coefficient of variation was ~3%). However in a few cases,
i.e. Lake Lucerne stations 1 and 2, Lake Baldegg station 3, $^{210}Pb$ profiles scattered more than $^{137}Cs$. Thus, in Lake Baldegg, varve counting was used for additional verification, and showed good agreement with rates based on $^{210}Pb$ and $^{137}Cs$. Sedimentation rates based on $^{210}Pb$ were then used to calculate mass accumulation rates (MAR; $g\ m^{-2}\ yr^{-1}$).

*Modeling of OC burial and accumulation rates through time.* The first-order decay of TOC was calculated using the power model of Middelburg (1989), which is a function of TOC % at the sediment surface, sediment age, and a location-specific,
empirically derived decay constant *k*. We then compared measured TOC % values to TOC % predicted by the decay curve




(TOC$_{modeled}$) to investigate past changes in TOC % accumulation and burial. Accordingly, subsurface TOC % values that are higher (lower) than the model curve indicate that more (less) OC was deposited at the time that the sediment layer was formed than today. We then calculate TOC burial rate, defined as g TOC buried per m$^2$ per year in any horizon below the top 2 cm of sediment, and used the model to calculate TOC accumulation rate, defined as g TOC deposited per m$^2$ per year to the sediment

5    surface (top 2 cm of sediment).

We calculate TOC burial as follows:

$$TOC\ burial\ rate\ (g\ C\ m^{-2}yr^{-1}) = TOC_{measured}\ (wt\ \%) * MAR\ (g\ m^{-2}yr^{-1}) \qquad \textbf{\textit{(1)}}$$

$$MAR\ (g\ cm^{-2}yr^{-1}) = -1 * slope\ (g\ cm^{-2}) * decay\ constant\ {}^{210}Pb\ (yr^{-1}) \qquad \textbf{\textit{(2)}}$$

$$dry\ mass\ (g\ cm^{-2}) = dw\ (g) * bulk\ density\ (g\ cm^{-3}) * \frac{thickness\ (cm)}{ww\ (g)} \qquad \textbf{\textit{(3)}}$$

Here the *slope* is the change of ln $^{210}$Pb (Bq g$^{-1}$) vs. the cumulative dry mass (g cm$^{-2}$) and *dw* and *ww* refer to sediment dry and wet weight, respectively. *Thickness* is the vertical thickness of each sampled sediment interval.

       Based on the decay constant of the power model, we back-calculated TOC loss within each sediment layer since its

10    deposition to the sediment surface. We then used this TOC loss to reconstruct the TOC content (TOC$_{reconstructed}$) and TOC accumulation rates that each sediment layer experienced historically when it was located at the sediment surface as follows:

$$TOC_{reconstructed} = TOC_{measured}\ (wt\ \%) + \left( \frac{(TOC\ loss\ (\%) * TOC_{measured}\ (wt\ \%))}{100} \right) \qquad \textbf{\textit{(4)}}$$

$$TOC\ accumul. rates\ (g\ C\ m^{-2}yr^{-1}) = TOC_{reconstructed}\ (wt\ \%) * MAR\ (g\ cm^{-2}yr^{-1}) \qquad \textbf{\textit{(5)}}$$

***Modeling of respiration rates.*** We investigated the relationship between trophic state and (a) OC mineralization rates, which we refer to as total respiration rates (RR$_{total}$), (b) DIC production rates (RR$_{DIC}$), and methanogenesis rates (RR$_{CH4}$) based on

15    DIC and CH$_4$ concentration profiles, whereby RR$_{total}$ is defined as the sum of RR$_{DIC}$ and RR$_{CH4}$. DIC profiles were corrected for dissolution and precipitation of calcium carbonate using the Ca$^{2+}$ porewater profiles (Fig. S5; measured during same run on IC as NH$_4^+$) (Hyun et al., 2017). Profiles of CH$_4$ and DIC were modelled using a power function with the formula

$$C_{cal} = C_0 + a * z^b \qquad \textbf{\textit{(6)}}$$





where $C_{cal}$ is the calculated concentration of the model output, $C_0$ is the concentration of DIC at the sediment surface (0–1 cm) or of $CH_4$ in the shallowest sample of the methanogenesis zone, $z$ is the depth in cm, and $a$ and $b$ are variables shaping the curve. The modeled concentrations where used to calculate a slope (mM cm$^{-1}$), which was calculated as follows:

$$slope = a * b * z^{(b-1)} \qquad (7)$$

From this slope the *flux (f)* was calculated:

$$f = -slope * D_{eff} * n \qquad (8)$$

5 where $D_{eff}$ is the effective diffusion of DIC or $CH_4$ in water. The effective diffusion at a specific temperature was corrected for the formation factor according to the formula:

$$D_{eff} = \frac{D}{T^2} \qquad (9)$$

where D is the diffusion coefficient and $T^2$ is a factor calculated by multiplying porosity with the *formation factor (F)*. This formation factor was calculated using the formula:

$$F = 1.02 * (n^{-1.81}) \qquad (10)$$

where $n$ is porosity, *1.02* is a unity factor (close to 1 means unity, i.e. same size and shape of particles) and 1.81 is an empirical 10 factor. Based on a study in Lake Zug from Maerki et al. (2004), the unity factor was set to 1.02 and the empirical factor to 1.81. Diffusion coefficients (D) as a function of temperature were calculated as $D_{CH4}$ (*10$^{-5}$ cm$^2$ s$^{-1}$) = 0.0439 * T (C°) + 0.76 for $CH_4$ (Witherspoon and Saraf, 1965; Gruca-Rokosz, 2018), and as $D_{HCO3-}$ (*10$^{-5}$ cm$^2$ s$^{-1}$) = 0.0002 * T$^2$ (C°) + 0.0172 T (C°) + 0.5463 for DIC (Zeebe, 2011).

In addition to determining RR$_{total}$, RR$_{DIC}$ , and RR$_{CH4}$, we calculated the ratio of DIC to $CH_4$ production rates 15 (RR$_{DIC}$:RR$_{CH4}$) as a proxy for the contribution of methanogenesis to total respiration. Furthermore, we divided RR$_{total}$ by the cell counts in each depth layer to estimate mean cell-specific respiration rates (RR$_{cell}$).

## 3 Results

In the first part of this section we document the impact of eutrophication, driven by water column P concentrations, on OC burial. In the second part we examine how these effects cascade further to alter respiration rates and distributions of respiration 20 pathways in sediments.



### 3.1 Basic Sediment Parameters

An overview of the sedimentation rates, lamination intervals, bottom water temperatures, sediment porewater pH, and TOC content at each station is shown in Table 2. Average sedimentation rates (in cm yr$^{-1}$, ±standard deviation) in the eutrophic lakes (Lake Greifen: 0.32±0.04; Lake Zug: 0.28±0.08; Lake Baldegg: 0.29±0.03) are slightly higher than in mesotrophic Lake Zurich

(0.23±0.03), and considerably higher than in oligotrophic Lake Lucerne: (0.13±0.06). Yearly laminations, indicating absence of significant macrofaunal sediment mixing, are present in several depth intervals at stations in Lake Greifen, Lake Baldegg, and Lake Zurich, but are absent from all stations in Lake Lucerne and Lake Zug (also see core photos in Fig. S3). These laminations mainly occur from the mid-upper to mid-bottom half of cores. The deep station of Lake Baldegg is an exception, in that laminations occur all the way to the bottom of the core. Bottom water temperatures at the time of sampling were between

5–9 °C. The pH in all sediment cores was close to neutral, and the average TOC contents were similar across all lakes, showing no relationship with trophic state.

### 3.2 Relationships between TOC and sediment depth

The TOC content in all five lakes ranges mostly from 2–4 % (total range: 1.1–5.3 %), with all stations showing a decrease with sediment depth (Fig. 2, Table 2). Despite this decrease with depth, most lakes show local subsurface peaks between 10–

30 cm depth, in particular the three eutrophic lakes. While TOC values between lakes show strong overlaps in the upper halves of cores, low TOC in deeper sediment layers suggests that historically two of the lakes, Lake Greifen and Lake Zug, were oligotrophic. This is confirmed by core images, which indicate that the deeper parts of the Lake Greifen and Lake Zug cores are dominated by organic-poor, calcium carbonate-rich clay ("Seekreide") (Fig. S3).

TOC$_{modeled}$ profiles, based on first-order decay of OC that was fitted to TOC % in surface sediments, compare to TOC

% as follows (Fig. 2): in the three eutrophic lakes, the model (locally) underestimates TOC % in the top 10–30 cm of cores, but matches or overestimates TOC % data in lower parts of cores. In Lake Zurich, the model matches the measured data well, with only slight local underestimations of TOC % in the upper halves of all three stations. In Lake Lucerne, the model generally matches TOC % in the top ~15 cm, but consistently overestimates TOC % in deeper layers.

TOC$_{reconstructed}$ data indicate that sediment layers of the three eutrophic lakes that are at depths of 10–30 cm today in

many cases received higher inputs of OC when they were located at the sediment surface than today's sediment surface. By contrast, the TOC$_{reconstructed}$ data in Lake Zurich and Lake Lucerne do not indicate past periods of consistently higher TOC input to surface sediments than today. Yet, irrespective of today's trophic state, the TOC$_{reconstructed}$ from all lake stations suggest that, on average, the TOC % at the sediment surface was lowest at the time that the deepest (and oldest) layers of sediment were deposited.



### 3.3 TOC burial and accumulation rates over time in relation to P concentrations

The correlation between trophic state and TOC burial becomes more clear when TOC values are plotted against time (Fig. 3). Reflecting the higher sedimentation rates and similar TOC % (Table 2), TOC burial rates increase with trophic state, being highest in the eutrophic Lake Baldegg and Lake Greifen and lowest in oligotrophic Lake Lucerne (Fig. 3). Reflecting the depth

trends in TOC %, TOC burial rates overall decrease with sediment age but show distinct peaks in layers deposited from 1940–1990. The reconstructed TOC accumulation rates support this observation and suggest that peak TOC accumulation rates occurred between 1960–1980 in the eutrophic lakes. In addition, two stations in Lake Baldegg (21 and 45 m water depth) have a later peak in TOC accumulation rate that dates to the turn of the 21$^{st}$ century.

In Lake Greifen and Lake Zug increases in TOC burial and accumulation rate took place later than in Lake Baldegg,

occurring mainly from 1920 onward, whereas in Lake Baldegg TOC burial and TOC accumulation rates were already high in the late 19$^{th}$ century. Incorporating changes in water column P concentrations from Fig. 1 into Fig. 3 shows that, in the eutrophic Lake Greifen, Lake Baldegg, and Lake Zug, increases in water column P concentrations occurred during the same time that TOC burial and accumulation rates increased during the 20th century. Peak P concentrations were measured between 1960 and 1980, when also the highest TOC accumulation rates were determined. Recent, strong decreases in water column P

concentrations match significant, albeit less marked decreases in TOC accumulation rates in Lake Greifen and Lake Zug. Comparing the time period of peak P concentrations to the time period from 2000-2014, P concentrations decrease by ~87 and 49% in Lake Greifen and Lake Zug, whereas average TOC accumulation rates only decrease by 24±11% and 22±5%, respectively. By contrast, in Lake Baldegg, where the decrease in P concentrations was the greatest during this time (~91%), there was no significant decrease in TOC accumulation rate (6±14%). In contrast to the eutrophic lakes, TOC accumulation

rates in Lake Zurich and Lake Lucerne show no clear response to the (minor) temporary increases in water column P concentrations during the 20$^{th}$ century. Similarly, though P concentrations have decreased by ~75% (Lake Zurich) and ~89% (Lake Lucerne) between the time of the P concentration peak and 2000-2014, TOC accumulation rates have not changed significantly (Lake Zurich: 0±4%) or in fact increased, albeit insignificantly (Lake Lucerne: 11±17%).

Correlation analyses suggest that across all five lakes there is a highly significant ($p<0.001$), positive correlation

between TOC accumulation rates and P concentrations (Fig. 4). For the entire time period from 1840–2016 considerable scatter and a low coefficient of determination ($R^2=0.28$) indicate that P concentrations only account for a minor portion of the observed variation in TOC accumulation rates (Fig. 4a). This changes, however, if only data from until the P peak are taken into account. Pairing these TOC accumulation rates with corresponding modeled and measured P concentrations (Fig. 4b) increases the $R^2$ to 0.54. When only TOC accumulation rates and measured P concentrations until the P peak are plotted (Fig. 4c), this $R^2$

increases further to 0.72. By contrast, the scatter increases ($R^2=0.23$), and the slope of the trendline decreases, when only TOC accumulation rates from after the P peak are correlated with measured P concentrations (Fig. 4a, d). Nonetheless, even during these recent decades, the correlation between TOC accumulation rates and P concentrations remains highly significant.



### 3.4 Distribution of microbial respiration reactions and microbial population size

*O$_2$.* At all stations, O$_2$ concentrations decrease in a consistent, concave down-fashion, indicating high rates of O$_2$ consumption in the absence of significant O$_2$ production (Fig. 5, inserts in first row). Average O$_2$ penetration depths are lowest in Lake Baldegg and Lake Greifen (Table 3), indicating highest rates of aerobic respiration. Penetration depths increase with less

eutrophic conditions in Lake Zug and Lake Zurich, and are greatest in Lake Lucerne, suggesting the lowest rates of aerobic respiration in the latter. Not surprisingly, the deep station from the hypoxic deep basin in Lake Zurich shows lower O$_2$ penetration than the shallower stations. In Lake Lucerne O$_2$ penetration increases from the shallow to the deep station. By contrast, the other three lakes show no trends related to water depth and are less variable in O$_2$ penetration depths.

*NO$_3^-$.* Bottom water concentrations of NO$_3^-$ are higher in Lake Greifen (57–100 µM) and Lake Baldegg (93–103 µM) than in Lake Zurich (29–45 µM), Lake Lucerne (39–45 µM), or Lake Zug (21–25 µM) (Fig. 5, first row). The wide ranges in bottom water NO$_3^-$ concentrations in Lake Greifen and Lake Zurich, which have lower NO$_3^-$ concentrations at the medium and deep station, and at the deep station compared to shallower stations, respectively, suggest denitrification in the (seasonally) hypoxic water column overlying these deeper stations. The NO$_3^-$ penetration depths are lowest in Lake Greifen and Lake Baldegg. This

and the overall high concentrations of nitrate indicate that denitrification rates are highest in these eutrophic lakes. Bottom water NO$_3^-$ concentrations and NO$_3^-$ penetration depths indicate that denitrification rates in surface sediments are next highest in Lake Zurich, followed by Lake Zug and Lake Lucerne, both of which have similar depth distributions of NO$_3^-$. Five stations (Lake Greifen 24 and 33 m, Baldegg 45 m, Zug 35 m, Zurich 137 m) have concave-down concentration decreases indicative of mainly or exclusively NO$_3^-$ consumption. The other stations have similar concave-down profiles in deeper strata, but have

concave-up profiles at the sediment surface, mostly in the top 1–2 cm, consistent with microbial NO$_3^-$ production by nitrification in this layer.

*Mn$^{2+}$.* Dissolved Mn$^{2+}$ was absent in significant concentrations in bottom water of all lakes, except at the hypoxic deep station in Lake Zurich (35 µM), and the deepest stations in Lake Greifen (12 µM) and Lake Baldegg (6 µM) (Fig. 5, second row). At

all stations, [Mn$^{2+}$] increases in the upper sediment layers, suggesting Mn$^{2+}$ production by microbial Mn(IV) reduction. Microbial Mn reduction occurs at lower rates than microbial Fe reduction, as indicated by lower [Mn$^{2+}$] than [Fe$^{2+}$]. The only exception is the deep station in Lake Zurich where [Mn$^{2+}$] exceed [Fe$^{2+}$] throughout the upper 30 cm of sediment. In all lakes except Lake Zug porewater [Mn$^{2+}$] increase from shallow to deep stations, suggesting enhanced microbial Mn reduction rates from shallow to deep sediments. At most stations [Mn$^{2+}$] increases in a concave-down fashion with depth, indicating

continuous, albeit decreasing Mn$^{2+}$-reduction rates throughout the cores. Yet, reliable interpretations of the lower depth limit of Mn-reducing microbial activity are in some places confounded by local [Mn$^{2+}$]-peaks, by strong [Mn$^{2+}$] decreases, and/or near-linear increases in [Mn$^{2+}$] profiles in subsurface layers (deep stations in Lake Greifen, Lake Zurich, and Lake Lucerne,



all stations in Lake Zug). Therefore, the lower depth limits of Mn-reduction, as indicated in Fig. 5 and Table 3, are associated with considerable uncertainty.

***$Fe^{2+}$.*** In all lakes, $[Fe^{2+}]$ is low (<2 µM) or absent in bottom water and increases in porewater of surface sediment due to microbial Fe reduction (Fig. 5, third row). At all stations, the highest porewater $[Fe^{2+}]$ were present in the core bottoms,

suggesting that the Fe reduction zone extends beyond the depth sampled (Table 3). In the eutrophic lakes, $[Fe^{2+}]$ accumulates rapidly, reaching 200–400µM (Lake Baldegg), 60–100 µM (Lake Greifen), and 120–160 µM (Lake Zug) in the top 5 cm. Below, concentrations continue to increase, reaching 500–540µM (Lake Baldegg), 250–500µM (Lake Greifen), and 380–530 µM (Lake Zug) at the core bottom. An exception is the deep station in Lake Baldegg, where the $[Fe^{2+}]$ has a local subsurface minimum at ~16 cmblf. Compared to the eutrophic lakes, Lake Zurich and Lake Lucerne appear to have lower microbial Fe-

reducing activity, as evidenced by lower $[Fe^{2+}]$ concentration maxima (Lake Zurich: 95–310 µM; Lake Lucerne: 120–200 µM). Furthermore, Lake Zurich has divergent $[Fe^{2+}]$ profiles from all other lakes, having - within the upper 8cm - consistently low (< 4µM) concentrations (shallow and medium stations) or a near-linear concentration increase (deep station). Thus, porewater $[Fe^{2+}]$ profiles that clearly indicate Fe reduction are not established until depths >8 cm (also see 'Solid-phase Fe pools and EAC/EDC' in this section).

***$SO_4^{2-}$.*** Bottom water concentrations of $SO_4^{2-}$ differ significantly between lakes, but show no trend in relation to trophic state. The oligotrophic Lake Lucerne (150–170 µM), mesotrophic Lake Zurich (130–150 µM), and eutrophic Lake Greifen (130–140µM) have comparable concentrations, that clearly exceed those in the eutrophic Lake Baldegg (90–110 µM) and Lake Zug (48–53 µM) (Fig. 5, fourth row). At all stations, porewater $[SO_4^{2-}]$ decreases steeply from the sediment surface downward, indicating already high rates of microbial $[SO_4^{2-}]$ reduction in the surface sedimentary layer, but penetrates deeper into the

sediment than $NO_3^-$. $SO_4^{2-}$ penetration depths are shallowest in the eutrophic Lake Greifen and Lake Baldegg and deepest in oligotrophic Lake Lucerne (Table 3).

***$CH_4$.*** At all stations, $[CH_4]$ increase with depth to the core bottom suggesting microbial methanogenesis beyond the cored intervals (Fig. 5, fifth row; Table 3). The highest $[CH_4]$ are reached in sediments of Lakes Baldegg and Greifen (~8 and ~6 mM, respectively), and suggest higher methanogenesis rates than in sediments of the other lakes. Lake Zug reaches maximum

$[CH_4]$ of 2–4.5 mM at the core bottom. In Lake Zurich, the shallow and medium stations reach $[CH_4]$ maxima of 1.7 and 2.2 mM, respectively, whereas concentrations reach 4 mM at the deep station. In Lake Lucerne, $[CH_4]$ remain <1.5 mM, indicating the lowest methanogenesis rates among all five lakes. Strong increases in $[CH_4]$ and $\delta^{13}$C-DIC (*data not shown*) in the three eutrophic lakes and the deep station of Lake Zurich suggest that the methanogenesis zone already begins within the top 1 cm of sediment. By contrast, absence of significant $[CH_4]$ accumulation and decreases in $\delta^{13}$C-DIC (*data not shown*) indicate

minimal methanogenesis, and instead significant rates of methane oxidation, in the top 2–3 cm of all Lake Lucerne stations and the shallow and medium stations in Lake Zurich.



***Solid-phase bioavailable Fe pools and EAC/EDC.*** Bioavailable Fe(III) scatters between 0–5 mM in all lakes except in surface sediment of Lake Lucerne, where concentrations up to 25 mM were measured in a ~2 cm thick, oxidized surface layer (Fig. S6). Despite the absence of a clear decrease in bioavailable Fe(III), bioavailable Fe(II) increases with depth at all stations, from 5–10 mM at the surface to 30–50 mM in deeper layers, consistent with microbial reduction of Fe(III), that is apparently not

limited to the 'bioavailable' fraction, to the bottom of all cores (Fig. S6)**.** The steepest increase in Fe(II) typically occurs in the top ~5cm, suggesting that iron reduction rates are highest in surface sediments. The electrochemical measurements confirm the trends in bioavailable Fe(II) and Fe(III), with a greater EDC than EAC everywhere except in the oxidized surface layer of Lake Lucerne, strong scatter in the EAC throughout all cores, and a general increase in EDC with depth (Fig. S6).

### 3.5 Total and cell-specific microbial respiration activity

At all stations, DIC concentrations increase with depth in a typical concave-down fashion that indicates highest rates of total respiration in surface sediments, and a decrease in microbial respiration rates with depth (Fig. 6, first column).

Sediments from Lake Baldegg and Lake Greifen show the highest DIC concentrations, reaching 6–9 mM near the core bottom. With a decrease in trophic state, there is a decrease in DIC accumulation, indicative of lower rates of microbial respiration. Accordingly, DIC maximum concentrations in the core bottoms reach 5–6 mM in Lake Zug, 4–5 mM in Lake

Zurich, and 3–4 mM in Lake Lucerne. Similar trends in relation to trophic state are present for $NH_4^+$ concentrations, which are a proxy for the microbial breakdown of N-containing organic matter, such as amino acids and nucleic acids (Fig. S5).

Consistent with the DIC profiles, the modeled $RR_{total}$ decrease with depth in all 5 lakes (Fig. 6, second column). While $RR_{total}$ decreases by approximately two orders of magnitude in all lakes, sediment horizons from eutrophic lakes have higher average $RR_{total}$ than the same sediment depths in oligotrophic lakes. For instance, the mean $RR_{total}$ in the shallowest layers is

2.1 nmol C cm$^{-3}$ h$^{-1}$, 5.8 nmol C cm$^{-3}$ h$^{-1}$, and 1.6 nmol C cm$^{-3}$ h$^{-1}$ in Lake Greifen, Lake Baldegg, and Lake Zug, respectively, compared to ~1 nmol C cm$^{-3}$ h$^{-1}$ in both Lake Zurich and Lucerne. This difference is maintained all the way to the bottom of the cores, where values in Lake Greifen (0.05 nmol cm$^{-3}$ h$^{-1}$), Lake Baldegg (0.03 nmol cm$^{-3}$ h$^{-1}$), and Lake Zug (0.02 nmol cm$^{-3}$ h$^{-1}$) remain higher than in Lake Zurich and Lake Lucerne (both 0.01 cm$^{-3}$ h$^{-1}$).

The ratio of $RR_{DIC}:RR_{CH4}$ indicates a higher contribution of methanogenesis to $RR_{total}$ in the three eutrophic lakes

(Fig. 6, third column). This difference is most pronounced in surface sediments where mean $RR_{DIC}:RR_{CH4}$ values are 0.3 in Lake Greifen, 1.2 in Lake Baldegg, and 2.3 in Lake Zug, compared to 3.5 in Lake Zurich and 12 in Lake Lucerne. With increasing sediment depth, the mean $RR_{DIC}:RR_{CH4}$ become more similar between the lakes. In Lake Greifen, the $RR_{DIC}:RR_{CH4}$ increases slightly with depth, indicating a decrease in the contribution of methanogenesis to $RR_{total}$. In all other lakes, the $RR_{DIC}:RR_{CH4}$ decreases with depth, indicating an increased contribution of methanogenesis to total respiration. Overall, the

$RR_{DIC}:RR_{CH4}$ change less with depth and are less variable between stations going from the eutrophic to the oligotrophic lakes.

Microbial cell numbers decrease with depth, showing similar general trends in all 5 lakes. Cell counts in surface sediments range from $10^9$–$10^{10}$ cm$^{-3}$ and decrease to ~$10^7$–$10^9$ cm$^{-3}$ in deeper layers (Fig. 6, forth column). Cell numbers, however, also show considerable scatter between stations (especially Lake Baldegg and Lake Zurich) and abrupt changes





between adjacent sediment intervals (especially Lake Greifen, Lake Baldegg, and the deep station in Lake Zurich). The abrupt drop in cell concentrations in the lower half of the core from the deep station in Lake Zurich matches a lithological shift toward organic-poor, carbonate clay ("Seekreide") layers, which are turbidites coming from upslope of the lake (Fig. S3). By comparison, the fluctuations in cell counts in Lake Baldegg and Lake Greifen do not match any measured geochemical or

sedimentological parameters. Microbial population size does not reflect trophic state or $RR_{total}$, as cell numbers strongly overlap between lakes. No clear subsurface peaks in cell counts, e.g. as relic signals of periods of peak eutrophication or TOC accumulation, are evident either. On the contrary, the highest average cell counts in surface sediments are from mesotrophic Lake Zurich and oligotrophic Lake Lucerne, and the highest average cell counts in deeper layers are from Lake Zurich (Fig. S7).

Reflecting the high variability in cell count data, the $RR_{cell}$ also have high variability. Nonetheless, there are several clear trends. In all lakes, $RR_{cell}$ are highest at the sediment surface and then decrease, stabilizing within the top 5–10 cm, and showing only minor changes throughout the remainder of the cores. Though the mean $RR_{cell}$ of surface sediment is not a good predictor of trophic state, with Lake Zurich having the highest $RR_{cell}$ (0.41 pmol C cell$^{-1}$ yr$^{-1}$), the $RR_{cell}$ reflect trophic state below 5–10 cm depth (Lake Greifen: ~0.005 pmol C cell$^{-1}$ yr$^{-1}$; Lake Baldegg: ~0.01 pmol C cell$^{-1}$ yr$^{-1}$; Lake Zug: ~0.004 pmol

C cell$^{-1}$ yr$^{-1}$; Lake Zurich: ~0.002 pmol C cell$^{-1}$ yr$^{-1}$; Lake Lucerne: ~0.0009 pmol C cell$^{-1}$ yr$^{-1}$). Based on a Paired Samples Wilcoxon Test, the average $RR_{cell}$ are not significantly different between Lake Baldegg and Lake Greifen, but are significantly different between Lake Baldegg and Lake Zug (p=0.04), Lake Zug and Lake Zurich (p=0.003), and Lake Zurich and Lake Lucerne (p=0.02).

### 3.6 *Relationships of $RR_{total}$, cell counts, and $RR_{cell}$ with sediment age*

Due to the clear differences in sedimentation rates between and within the lakes, analyzing $RR_{total}$, microbial population size, and $RR_{cell}$ in the context of sediment depth alone does not provide reliable insights into the impacts of eutrophication. We therefore subdivided the study period into 6 time intervals, the pre-eutrophication era (1840–1900), three time intervals of the eutrophication era (early: 1900–1940, mid: 1940–1960, peak: 1960–1980), and two time intervals from the era after rigorous wastewater treatment had been established and during which P bans on detergents, chemical P precipitation systems in

wastewater treatment, and artificial mixing and aeration of lake water columns were implemented (1980–2000, 2000–2016). Within each time interval, we averaged the data from each station (Fig. 7). Due to the high intra-lake and intra-core variability, the standard deviations are high and in many cases overlap. Nonetheless, several insights into the relationships between eutrophication, microbial population size, and microbial respiration rates can be gained.

In all lakes, $RR_{total}$ decreases with sediment age in a near-asymptotic way, with a one order of magnitude decrease

across sediment intervals from the last ~60 years, followed by minor decreases below (Fig. 7a). Thus, sediments deposited during the pre-eutrophication, early and mid eutrophication periods have similar $RR_{total}$ today, even though TOC accumulation rates changed significantly across these periods (Fig. 3).



Cell numbers decrease with sediment age (Fig. 7b) confirming the earlier observation that eutrophic lakes do not have higher cell counts than oligotrophic lakes (Fig. 6). When plotted against age since 1900, Lake Lucerne and Lake Zurich even have consistently higher average microbial abundances than the three eutrophic lakes. Depth-related decreases in cell numbers are also different between the lakes. The two lakes that went from highly oligotrophic to eutrophic (Lake Greifen, Lake Zug) have ~20fold (Lake Greifen) and ~30fold (Lake Zug) decreases in average cell numbers from 2000–2016 to 1840–1900. By comparison, cell counts in the other three lakes have smaller changes in cell numbers with sediment age (Lake Baldegg: ~4fold; Lake Zurich: ~8fold; Lake Lucerne: ~6fold).

The $RR_{cell}$ decrease within the first ~4 decades since deposition, but stabilize in sediments that were deposited before 1980 (Fig. 7c). The only exception is Lake Baldegg, where the $RR_{cell}$ remains stable throughout all layers deposited since 1940, and only decreases in horizons deposited prior to 1940. The decrease in $RR_{cell}$ over time since deposition is approximately one order of magnitude in Lakes Greifen, Baldegg, Zug, and Lucerne, and two orders of magnitude in Lake Zurich.

## 4 Discussion

Our results show that eutrophication of lakes in central Switzerland, driven by anthropogenic input of P, has had a strong impact not only on water columns but also on sediments. While the average TOC content is not a good indicator of trophic state (Table 2), changes in $TOC_{reconstructed}$ with depth indicate a significant impact of eutrophication on the sedimentary TOC sink (Fig. 2) with significant increases in TOC burial and accumulation rates over the last century (Fig. 3). Correlations between TOC accumulation rates and water column P concentrations through time (Fig. 3) suggest that lakes with the highest increases in P concentrations also had the highest increases in TOC accumulation and burial rates in their sediments. The relationship between P concentrations and TOC accumulation rates was strongest during the period of rapid P concentration increase in the mid 20$^{th}$ century (Fig. 4). Yet, even though water column P concentrations in most lakes have decreased close to pre-eutrophication levels since the ~1970s, TOC burial and accumulation rates in eutrophic lakes remain significantly higher than before the eutrophication era (Figs. 3 and 4).

Despite the increase in TOC burial, lake sediments are not a static sink for OC. Increases in TOC accumulation and burial increase remineralization by stimulating microbial respiration (Fig. 6). Water column-derived electron acceptors, such as $O_2$, nitrate, and sulfate, are depleted at shallower depths in eutrophic lakes compared to lakes with a lower trophic status (Fig. 5; Table 3). Similarly, the contribution of methanogenesis to total respiration is greater (Fig. 6), and significant rates of methanogenesis occur at shallower sediment depths in the eutrophic lakes (Fig. 5, Table 3). These eutrophication-related changes in respiration rates are, however, not reflected in the vertical zonation of microbial respiration pathways with respect to each other. Instead, the depth intervals of denitrification, Mn(IV) reduction, Fe(III) reduction, sulfate reduction, and methanogenesis show strong overlaps, independent of lake trophic state (Fig. 5; Table 3). Also, although higher TOC input and respiration rates appear to increase cell-specific respiration rates, they do not result in a higher microbial population size



(Fig. 6). Instead, the lakes that were least affected by eutrophication (Lake Zurich, Lake Lucerne) have the highest cell counts in sediments deposited during the period of peak eutrophication from 1940–2000 (Fig. 7).

In the following sections, we discuss in more detail the *(i)* response of TOC accumulation and burial to eutrophication, *(ii)* potential effects of eutrophication mitigation measures on TOC burial and accumulation, *(iii)* the relationship between zonation and rates of dominant respiration processes and trophic state, and *(iv)* the relationship between total respiration rates, microbial population size, and trophic state.

### *(i)*    *Response of TOC burial to eutrophication*

The TOC burial rates in this study match estimates from previous studies. The average annual burial rate of 27 g C m$^{-2}$ yr$^{-1}$ for sediments deposited from 1960 to 1990 at the deep station of Lake Baldegg is only slightly lower than the estimate of Teranes and Bernasconi (2000) (~30 g C m$^{-2}$ yr$^{-1}$) for the same location and time interval. A recent study on Lake Baldegg determined average OC burial rates of 32.3 g C m$^{-2}$ yr$^{-1}$ for the top 2–10 cm of stations from water depths of 23, 40, 64 m (Steinsberger et al., 2017), which are very similar to the  34 g C m$^{-2}$ yr$^{-1}$ for the top 2–10 cm of our three 3 stations. The average TOC burial rates from our study (13.8 to 33.3 g C m$^{-2}$ yr$^{-1}$ from 1840 to 2016), moreover, fall within the range of 116 lakes from different ecoregions in Minnesota (3 to 184 g C m$^{-2}$ yr$^{-1}$ from 1800–2010) (Anderson et al., 2013), and from 93 lakes in 11 countries across Europe (Anderson et al., 2014). The latter study, estimated that average burial rates increased on average 2.5-fold from 17 g C m$^{-2}$ yr$^{-1}$ in the 19$^{th}$ century to 40 g C m$^{-2}$ yr$^{-1}$ in 1900–1950, and again 1.5-fold to 60 g C m$^{-2}$ yr$^{-1}$ after 1950 (Anderson et al., 2014). By comparison, we estimate a smaller impact on TOC burial in the 5 lakes studied. Our average±SD burial rates (in g C m$^{-2}$ yr$^{-1}$) of 14±6 (1900–1940), 17±6 (1840–1900), 22±8 (1940–1960), 27±10 (1960–1980), 27±9 (1980–2000), and 29±8 (2000–2016) only correspond to a doubling in average TOC burial rates from the 19$^{th}$ century to after 1960. Incorporating estimates of TOC loss over time does not significantly change these interpretations. According to the power model (Middelburg, 1989), the biggest TOC loss occurs in year 1 after deposition (~16%), followed by ~2% over the following 15 years (in our case 2000–2015), ~1% over the following 20 years (1980–2000), and ~1.5% over the following 140 years (1840–1980). Our burial rates and those determined in the other above studies were all determined on sediments that were over 1 year old.

While our data indicate that TOC burial and accumulation rates increase with eutrophic conditions, it is important to distinguish between natural background OC accumulation rates, and increases in TOC accumulation rates above these background rates that are due to eutrophication. For instance, in Lake Greifen and Lake Baldegg, TOC accumulation rates were already 2–3 times higher than in Lake Lucerne before the period of eutrophication (Fig. 3). Lake Zug and Lake Zurich also had higher average TOC accumulation rates than Lake Lucerne prior to eutrophication. Variables besides anthropogenic eutrophication, e.g. natural OM inputs from the watershed or surrounding riparian vegetation, and physical factors, such as sediment focusing, lake area, water depth, and water residence times must contribute to these natural differences between the five lakes. While we do not have data on OM contributions from land or riparian zones, physical factors may indeed (partially)



drive the natural differences in TOC accumulation rates between the five lakes (Fig. S8 and Table S1). In line with previous studies (Dean and Gorham, 1998; Heathcote and Downing, 2012), we observe a strong negative linear relationship between lake area and TOC accumulation rate (Fig. S8a; $R^2$=0.99). This may be (partially) due to dilution effects, as lake volume also shows a strong negative linear relationship with TOC accumulation rates (Fig. S8b; $R^2$=0.91). By contrast, water residence

time shows no correlation with TOC accumulation rates (Fig. S8c). Other variables that often drive the distribution of TOC accumulation within lakes, e.g. water depth and sediment focusing (Lehman, 1975; Davis and Ford, 1982; Blais and Kalff, 1995), also show no clear trends in relation to TOC accumulation rates (Fig. S8d–e) and SI Text 2.

In spite of these pre-existing natural differences in lake-specific TOC accumulation rates, the clear changes in TOC accumulation (and burial) rates over time, which correlate with water column P concentrations and coincide with anoxic events

and algal blooms (Figs. 1, 3, and 4), indicate a strong impact of eutrophication on lake TOC burial. Similar increases in OC burial and accumulation in lake sediments due to eutrophication have been reported elsewhere (Gorham et al., 1974; Dean and Gorham, 1998; Heathcote and Downing, 2012). A combination of variables may drive this eutrophication-related increase in OC burial. First, the increase in primary production due to lake P fertilization promotes water column biomass production, and increases biomass deposition to the lake floor. Increased OM loading then stimulates aerobic OM catabolism, which causes

water column anoxia, which further promotes OM preservation and burial (Lehmann et al., 2002; Sobek et al., 2009; Katsev and Crowe, 2015).

### *(ii)       Effects of eutrophication mitigation measures on TOC burial*

Since the period of peak eutrophication, water column P concentrations have decreased back to pre-eutrophication levels, mainly as a result of wastewater treatment combined with P precipitation systems and removal of phosphate from detergents

(Fig. 1, SI Text 1). This coincided with significant decreases in TOC accumulation rates in Lake Greifen (24±11%) and Lake Zug (22±5%), but not in Lake Baldegg (6±14%; *discussed later in this section*). Presumably, the decreases in Lake Greifen and Lake Zug were mainly driven by reduced primary production. Decreased TOC preservation due to increased $O_2$ exposure time was probably less important, since seasonal water column anoxia continued for decades after the decrease in TOC accumulation rates in Lake Greifen and Lake Zug (Figs. 1 and 3).

Despite the observed decreases in TOC accumulations in two of the three eutrophic lakes, our calculated TOC accumulation rates for the period after peak eutrophication have remained well above those during pre-eutrophication times (Fig. 3). Furthermore, the correlation between P concentrations and TOC accumulation rates has become weaker since the decrease in P concentrations (Fig. 4b–d). A potential reason for the reduced coupling between P concentrations and TOC accumulation rates is remobilization of P, which accumulated in sediments in detrital form during periods of high P inputs,

back into overlying water (Rippey and Anderson, 1996; Boyle, 2001; Meals et al., 2010; Giles et al., 2016). As known from other lakes, remobilization of P from sediments can sustain high primary productivity and lower water quality for years to decades after substantial decreases in P inputs (Lotter, 2001; Giles et al., 2016). Such remobilization of P is not necessarily



evident based on total P concentrations as most remobilized P is present in highly bioavailable inorganic form that is released by microbial remineralization of sedimentary OM. By contrast, most P from wastewater or manure is present in organic form and has to be remineralized to inorganic P species by microbial heterotrophs to become available to primary producers. This slower turnover of organic P from wastewater may in part explain why P concentrations soared during the period of peak

eutrophication, but are low today, despite the significant release of remobilized P from sediments.

Our data, furthermore, indicate that artificial mixing does not substantially reduce TOC accumulation and burial, and in fact might counteract reductions in TOC accumulation and burial caused by reduced P inputs. In Lake Baldegg, artificial aeration and oxygenation were implemented in 1982/3 (Stadelmann and Escher, 2002) and have eliminated bottom water anoxia, thereby increasing $O_2$ exposure time of most detrital OC from hours, during sinking through the water column (Bloesch

and Burns, 1980; Lehmann et al., 2002), by an additional 3–4 months within oxic sediment [note: $O_2$ exposure time in sediment was estimated by dividing $O_2$ penetration depth by sedimentation rate]. Nonetheless, TOC accumulation and burial rates have not decreased significantly since then, confirming initial observations made by Gächter and Wehrli (1998) after 10 years of artificial aeration. Given that Lake Greifen and Lake Zug, which were not artificially mixed, had significant reductions in TOC accumulation rates, we speculate that water column mixing may even be part of the reason for the continuing high TOC burial

rates in Lake Baldegg. Accordingly, mixing creates an "artificial upwelling" that efficiently transports remobilized P, which has diffused from sediments into bottom water, to the photic zone. In the photic zone, this P is turned over rapidly and sustains high growth rates of photosynthetic organisms. High rates of primary production then sustain the continually high rates of TOC accumulation and burial in lake sediments. If this interpretation is correct, then we would predict TOC burial and accumulation rates in Lake Greifen to remain stable or increase since the beginning of artificial mixing in 2009. Comparing

TOC accumulation rate data from 2000-2009 to data from 2010-2014 suggests this could be the case, as TOC accumulation rates have increased, albeit variably, at all three stations since then by on average 6±5%.

### (iii)    *Zonation and rates of dominant respiration processes*

The vertical distribution of dominant respiration reactions in the five lakes generally matches what would be expected based on differences in total respiration rates. Electron acceptors that diffuse from bottom water into sediment ($O_2$, nitrate, sulfate)

are depleted at shallower depths in eutrophic compared to the meso- and oligotrophic lakes due to higher respiration rates fueled by higher TOC input. Less reactive solid-phase electron acceptors (Mn(IV), Fe(III)), and electron acceptors that are produced at high rates within sediments ($CO_2$), extend deeper into cores, and even throughout cores in the case of Fe(III) and $CO_2$. While Mn reduction rates appear to be mainly controlled by Mn supply, which is driven by geochemical focusing (Engstrom and Wright Jr, 1984; Schaller and Wehrli, 1996; Naeher et al., 2013), both Fe reduction and methanogenesis rates

increase with trophic state, most likely as a result of increased electron donor and acceptor supply from the enhanced breakdown of OM.





In addition to this general effect of eutrophication on microbial respiration rates, a clear vertical separation of respiration reactions involving different electron acceptors is absent, independent of trophic state (Fig. 5). One possible reason is that electron donor concentrations are not under thermodynamic control (Hoehler et al., 1998), and that free energy yields of all respiration reactions are above previously observed minimum thresholds per reaction (10 to 20 kJ mol$^{-1}$; Hoehler et al. (2001)). This could potentially indicate that respiration reactions in sediments were not in steady-state at the time of sampling, e.g. due to temperature changes, redox fluctuations, mortality, and/or bioturbation (Hoehler et al., 1999; Lever and Teske, 2015; Chen et al., 2017). Chemical microenvironments with divergent redox conditions may also enable different respiration reactions to occur physically separated but in close proximity to each other within the same sediment horizon (Anderson and Meadows, 1978; Canfield, 1989; Oda et al., 2008). "Non-competitive" substrates, such as the C1 compounds methanol, dimethylsulfide, and trimethylamine, which are consumed by some methanogens and not by most sulfate reducers, and enable coexistence of these methanogens with sulfate reducers in sulfate reducing marine sediment (Oremland and Polcin, 1982; Xiao et al., 2017) are also possible. However, previous research on lake sediments indicates that non-competitive substrates only support a small fraction of total methanogenesis (Conrad et al., 2011; Liu et al., 2017). Moreover, research on the five lakes studied here suggests dominance of $H_2/CO_2$-consuming *Methanomicrobiales*, and that methanogenesis from $H_2/CO_2$ is thermodynamically not favorable in most methanogenic horizons (Michel & Lever, *unpubl. data*). Thus, microenvironments with divergent redox conditions may explain the strong spatial overlaps in respiration reactions.

### (iv)    *Respiration rates and total microbial population size*

Total respiration rates (RR$_{total}$) decrease with sediment depth at all stations, indicating an increase in the OM degradation state over time, as microorganisms selectively mineralize the more reactive OM pools. The higher RR$_{total}$ in eutrophic compared to meso- and oligotrophic lakes, even in buried intervals that were deposited around the same time, suggests that the eutrophication-related stimulation of RR$_{total}$ persists over many decades, possibly due to the higher amounts of TOC (both reactive and non-reactive) that were initially deposited. Despite this continued OM remineralization, only a small fraction of originally accumulated TOC is remineralized in these older layers (also see final paragraph of Discussion section (i)).

Overall cell numbers in the range of $10^8$ to $10^{10}$ cells cm$^{-3}$ (Fig. 6 and S7) are in a similar range as other lake sediments (Bostrom et al., 1989; Haglund et al., 2003; Schwarz et al., 2007). Surprisingly, despite the higher TOC burial and RR$_{total}$, the three eutrophic lakes have lower average cell numbers than Lake Zurich and Lake Lucerne (Fig. 6 and S7). Possibly, the shallower depletion of $O_2$, nitrate, sulfate, and Mn(IV) in eutrophic lakes causes a larger fraction of respiring microorganisms to depend on energetically less favorable methanogenesis. Thus, cells in eutrophic lake sediment would have less energy available per cell despite having more available electron donors and higher RR$_{cell}$. Though possible, this scenario is unlikely given that the vast majority of anaerobic microorganisms in sediment are probably involved in fermentation reactions (Lever, 2013), and that methanogens account for at most a few percent of the microbial populations in cores from this study (Michel & Lever, *unpubl. data*). Alternatively, sediments of the eutrophic lakes may contain more toxic contaminants (Pritchard and Bourquin, 1984),




or have higher viral mortality (Fischer and Velimirov, 2002). Predation by oligochaete worms, which were abundant to sediment depths of >20cm in the eutrophic lakes (Fiskal et al., *unpubl.),* could also be important.

## 5    Conclusions

Based on sedimentary records from five lakes differing in eutrophication history over the last ~180 years, we demonstrate clear links between human activity on land, water column eutrophication, and OC burial in lake sediment. By combining high resolution historic data on water column P concentrations with reconstructed past TOC accumulation rates, we show that anthropogenic input of P to lake ecosystems, most likely by increasing water column primary production and TOC sedimentation, is a key driver of TOC input to and TOC burial within lake sediment. This relationship between P concentrations

in the water and TOC accumulation rates in sediments was strongest during the period leading to the eutrophication peak in the 1970s. Since then, despite reductions in P concentrations by 50-90% across all five lakes, TOC accumulation rates have decreased only by ~20-25% in two of the eutrophic lakes (Lake Greifen, Lake Zug), and not significantly in the third eutrophic (Lake Baldegg), a mesotrophic lake (Lake Zurich), and an oligotrophic lake (Lake Lucerne). This relatively small or absent decrease in TOC accumulation rates is most likely due to the efficient remobilization of sedimentary P by microbial diagenesis

and water column mixing. The fact that the highly eutrophic Lake Baldegg, which has been artificially mixed and aerated for >35 years to mitigate water column anoxia, shows no decrease in TOC accumulation rates or TOC burial, indicates that water column mixing and aeration is not effective at reducing P release from sediment or at controlling water column primary productivity, but might in fact have the opposite effect.

In addition to documenting the effects of eutrophication on TOC accumulation and burial, we investigate effects of

eutrophication on microbial TOC mineralization processes and microbial population size. We show that eutrophic lakes have higher respiration rates (total and cell-specific), and a higher contribution of methanogenesis to total respiration, than lakes with a lower trophic status. Yet, trophic state does not affect the zonation of microbial respiration reactions. Instead, independent of trophic state, low-energy reactions, such as methanogenesis, occur well into sediment layers with denitrification, manganese reduction, and even aerobic respiration, possibly due to the presence of geochemically distinct

microenvironments. Despite the higher respiration rates in the eutrophic lakes, we observe equal or even lower cell abundances in eutrophic compared to mesotrophic and oligotrophic lake sediment. This indicates that, for yet unknown reasons, electron donor supply and/or microbial energy availability is decoupled from microbial biomass in lake sediment.



**Data availability**

The data used in this manuscript will be made available on PANGAEA after the manuscript is published.

**Author contribution**

A.F., L.D., X.H., A.M., P.E., L.L., R.Z., N.D. and M.A.L. helped with sample collection, and/or measurements. A.F., N.D.,

M.H.S., S.B., M.S. and M.A.L substantially contributed to the interpretation of data. A.F. and M.A.L. wrote the manuscript.

M.A.L. designed the study and acquired the funding for the project. All authors commented on the manuscript and approved

the final version of the manuscript.

**Conflict of interest**

The authors declare no conflict of interests.

**Acknowledgements**

We thank Madalina Jaggi for sample preparation and measurements. Special thanks to Adrian Gilli (ETHZ) and all technicians, especially Irene Brunner, Alfred Lück and Alois Zwyssig from EAWAG for great help with sampling and downstream

analysis. Many thanks to Iso Christl, Rachele Ossola, Joep van Dijk and Andreas Brand for consulting regarding measurement and analyses. We thank Meret Aeppli and Nicolas Walpen for the introduction to electrochemical analyses. We thank Joachim Hürlimann for kindly providing reconstructed phosphorus data from Lake Greifen and Lake Zug and Andre Lotter for providing reconstructed phosphorus data for Lake Baldegg and Lake Lucerne. This project is funded by the Swiss National Science Foundation (project 205321_163371: "Role of bioturbation in controlling microbial community composition and

biogeochemical cycles in marine and lacustrine sediments" awarded to Mark A. Lever).




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





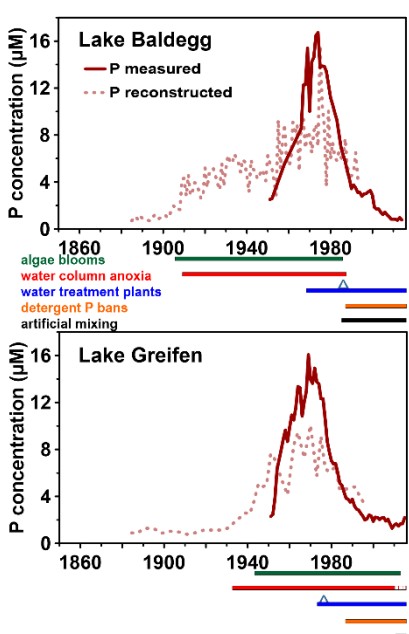

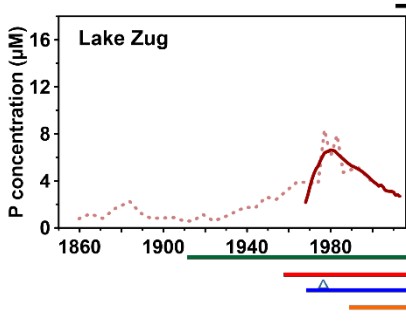

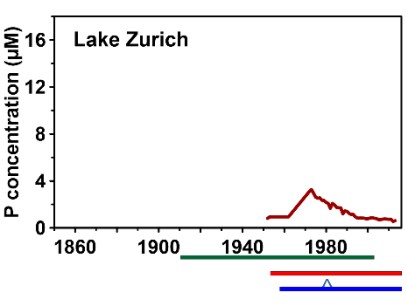

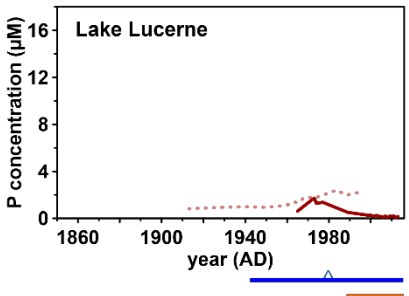

**Figure 1: Historical data on surface water P concentrations and eutrophication since 1860. P concentration monitoring was started between 1950 and 1970. Reconstructed P concentrations were modeled based on sedimentary records of diatom assemblages. Below each graph, timelines of algae bloom and anoxic event occurrence, wastewater treatment, and artificial mixing are shown. P bans were implemented in 1986 and are marked by blue triangles. All data from Swiss Federal Office of the Environment (BAFU 2011, 2013), except reconstructed P concentrations (Lake Lucerne: Lotter (2001); Lake Baldegg: Lotter (1998); Lake Zug; Elber et al. (2001); Lake Greifen: Elber et al. (2004); no data exist for Lake Zurich).**





**Figure 2: TOC (solid black line), TOC_modeled (solid green line), and TOC_reconstructed (grey dashed line). Water depth [m] for each station is indicated at the top of each subplot; cmblf = cm below lake floor.**





**Figure 3: TOC burial rates (black solid lines), reconstructed TOC accumulation rates (dashed grey lines), measured water column total P concentrations (red solid lines), and reconstructed water column total P concentrations (red-dotted lines) all plotted against sediment age at various lakes and lake stations.**



**Figure 4:** TOC accumulation rates [g C m⁻² yr⁻¹] vs. P concentrations for (a) the entire P dataset (1840-2016), (b) measured and reconstructed P concentrations until the P concentration peak in the 1960s and 1970s (includes the peak), (c) only measured P concentrations for the time period until the P concentration peak (includes the peak), and (d) measured P concentrations for the time period after the P concentration peak. N is the number of data points included in each graph. Coefficients of determination ($R^2$) values are based on best-fit Power trendlines. P-values are based on two-sided Spearman Rank Correlation tests. [Note: for Lake Zurich, only measured P data were available. Furthermore, in (a) and (b), for data points where both measured and reconstructed P concentrations were available, we only included measured values.]





**Figure 5: Concentration profiles of $O_2$ and $NO_3^-$, $Mn^{2+}$, $Fe^{2+}$, $SO_4^{2-}$, and $CH_4$ vs. sediment depth (cmblf) organized by analyte and lake. Each graph shows concentration profiles of all three stations per lake. Light brown horizontal bars indicate the depth of the sediment-water-interface. $CH_4$ measurements with significant outgassing during sampling were omitted. The inferred depth distributions of respiration reactions are indicated by vertical bars to the right of each graph. Notes: (1) depth ranges on y-axes vary between analytes; (2) analyte concentration ranges on x-axes vary with lakes and analytes.**



**Figure 6: Depth profiles of DIC concentrations, total microbial respiration rates ($RR_{total} = RR_{DIC} + RR_{CH4}$), ratios of DIC production to methanogenesis rates ($RR_{DIC}:RR_{CH4}$), cell abundances, and cell-specific respiration rates ($RR_{cell}$). DIC concentrations and cell counts are shown for each station individually. $RR_{total}$ and $RR_{total}:RR_{CH4}$ are lake-specific averages (black line) with standard deviation ranges (grey areas). A 1:1 line indicates $RR_{DIC} = RR_{CH4}$. $RR_{cell}$ are averages±SD for samples with cell count data.**





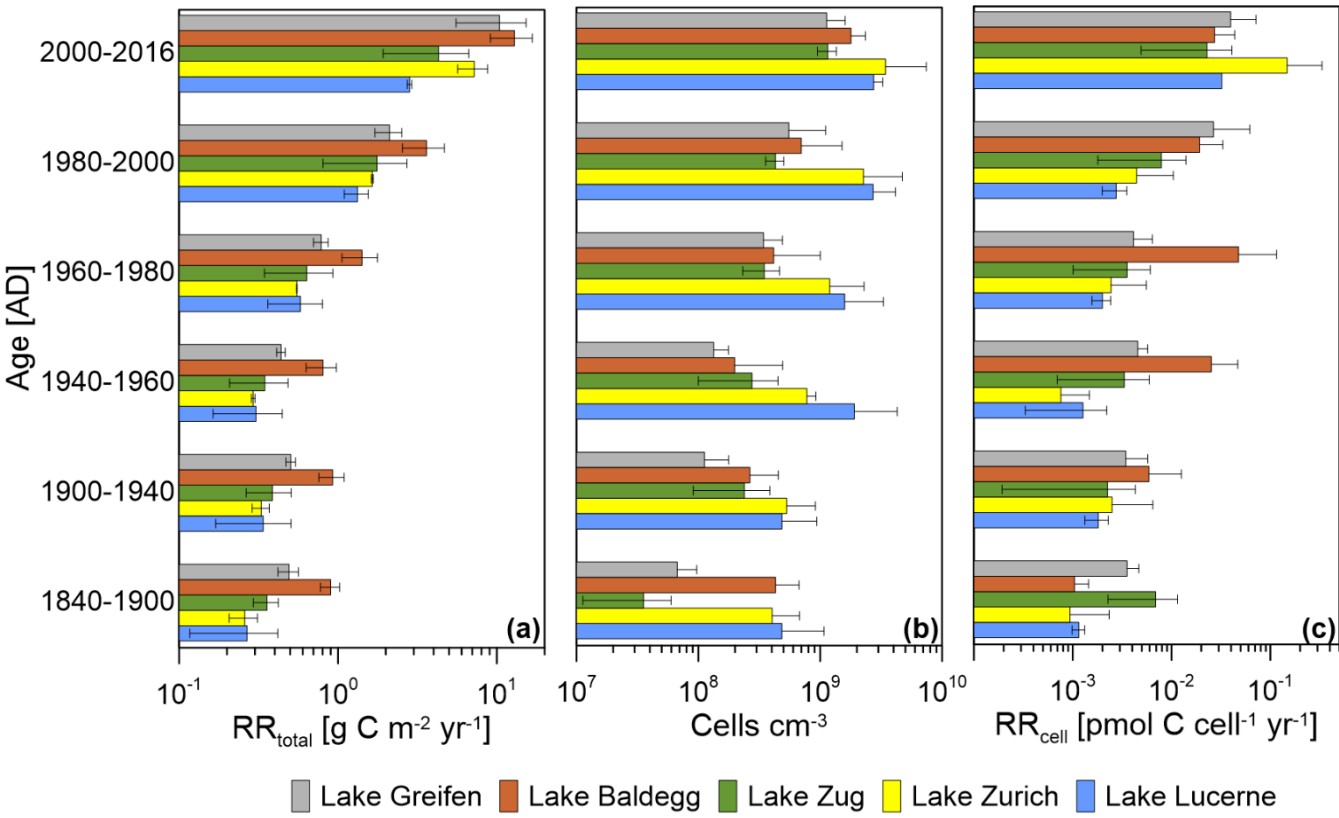

**Figure 7: Relationships between RR$_{total}$, microbial abundance, and RR$_{cell}$ and sediment age. Error bars indicate standard deviations of averages for the three stations per lake during each time interval.**



**Table 1: Overview of sampled lakes, their trophic status, and maximum water depths (from BAFU 2012, 2013), as well as the geographic coordinates, water depths, and bottom water dissolved O₂ concentrations of the stations that were sampled. O₂ concentrations ≤15.6 µM are termed 'hypoxic'.**

|  | Trophic status | Max. depth (m) | Station # | Latitude (°N) | Longitude (°E) | Water depth (m) | O₂ (µM) |
|---|---|---|---|---|---|---|---|
| **Lake Greifen** | eutrophic | 32 | 1 | 47° 21.134 | 8° 40.511 | 15 | seasonally hypoxic |
|  |  |  | 2 | 47° 21.118 | 8° 40.484 | 32 |  |
|  |  |  | 3 | 47° 21.038 | 8° 40.185 | 24 |  |
| **Lake Baldegg** | eutrophic | 66 | 1 | 47° 11.929 | 8° 15.613 | 66 | 15.6–125 |
|  |  |  | 2 | 47° 11.759 | 8° 15.392 | 45 | 125–250 |
|  |  |  | 3 | 47° 11.649 | 8° 15.417 | 21 | 15.6–125 |
| **Lake Zug** | eutrophic | 198 | 1 | 47° 10.272 | 8° 30.036 | 25 | 125–250 |
|  |  |  | 2 | 47° 10.104 | 8° 29.946 | 35 | 125–250 |
|  |  |  | 3 | 47° 09.834 | 8° 29.814 | 50 | 125–250 |
| **Lake Zurich** | mesotrophic | 137 | 1 | 47° 16.995 | 8° 35.624 | 137 | hypoxic |
|  |  |  | 2 | 47° 16.708 | 8° 35.033 | 45 | 125–250 |
|  |  |  | 3 | 47° 16.395 | 8° 35.195 | 25 | 15.6–125 |
| **Lake Lucerne** | oligotrophic | 214 | 1 | 47° 00.051 | 8° 20.218 | 24 | >250 |
|  |  |  | 2 | 46° 59.812 | 8° 20.820 | 93 | >250 |
|  |  |  | 3 | 46° 59.915 | 8° 20.413 | 45 | >250 |

**Table 2: Sedimentation rates, depth distributions of laminated layers, bottom water temperature, porewater pH, and average % TOC (± standard deviation (SD); 1920–2016) across stations in each lake. Triangles indicate laminations with the presence of turbidites. Stars indicate sediments without clear ¹³⁷Cs peaks. Sedimentation rates changed in Lake Zurich (25 m), from 0.23 cm yr⁻¹ in the top 7 cm to 0.27 cm yr⁻¹ below, and in Lake Lucerne (45 m) from 0.16 cm yr⁻¹ in the top 5 cm to 0.14 cm yr⁻¹ below.**

|  | Station (water depth in m) | Sed. rate (cm yr⁻¹) | Yearly lamination intervals (cm) | Temperature (°C) | PW pH ranges | TOC [wt %] |
|---|---|---|---|---|---|---|
| **Lake Greifen** | shallow (15 m) | 0.29 | 10–17 | 7 | 7.29–7.75 | 2.9 (± 0.8) |
|  | medium (24 m) | 0.31 | 2–21 | 7 | 7.18–7.88 | 2.5 (± 0.9) |
|  | deep (33 m ) | 0.37 | 2–35 | 6 | 7.21–7.46 | 3.1 (± 0.7) |
| **Lake Baldegg** | shallow (21 m) | 0.29 (*) | 11–13 | 7 | 7.37–8.38 | 3.0 (± 0.4) |
|  | medium (45 m) | 0.32 (*) | 20–30 | 7 | 7.35–8.30 | 2.2 (± 0.3) |
|  | deep (68 m) | 0.27 | 7–40 | 7 | 7.54–8.15 | 2.8 (± 0.3) |
| **Lake Zug** | shallow (25 m) | 0.22 | no | 7 | 7.46–7.82 | 3.0 (± 0.6) |
|  | medium (35 m) | 0.25 | no | 7 | 7.41–8.00 | 2.8 (± 0.5) |
|  | deep (50 m) | 0.37 | no | 7 | 7.37–8.02 | 3.1 (± 0.5) |
| **Lake Zurich** | shallow (25 m) | 0.23/0.27 | no | 7 | 7.38–8.07 | 2.7 (± 0.4) |
|  | medium (45 m) | 0.35 (*) | 20–35 | 7 | 7.50–8.08 | 2.7 (± 0.4) |
|  | deep (137 m) | 0.2 | 0–20ᐃ, 26–28ᐃ, 37–38ᐃ | 6 | 7.18–8.01 | 3.4 (± 0.6) |
| **Lake Lucerne** | shallow (24 m) | 0.17 | no | 9 | 7.40–7.70 | 3.4 (± 0.9) |
|  | medium (45 m) | 0.16/0.14 | no | 7 | 7.38–7.84 | 3.0 (± 0.4) |
|  | deep (93 m) | 0.06 | no | 5 | 7.42–7.84 | 2.9 (± 0.3) |





**Table 3: Average ± SD depth distributions (cmblf) of microbial respiration reactions in each lake.**

|  | **Greifen** | **Baldegg** | **Zug** | **Zurich** | **Lucerne** |
|---|---|---|---|---|---|
| **Aerobic** | surface – 0.17±0.03 | surface – 0.08±0.02 | surface – 0.23±0.03 | surface – 0.22±0.08 | surface – 0.73±0.25 |
| **Denitrification** | surface – 2.5±1.0 | surface – 2.8±1.2 | surface – 7.7±3.1 | surface – 3.3±1.4 | surface – 9±2.0 |
| **Mn reduction** | 0.3±0.3 – 8.2± 4.9 | 0.3±0.3 – 14.0±7.0 | 0.5±0.0 – 5.0±0.0 | 0.3±0.3 – 24.0±15.1 | 0.7±0.8 – bottom |
| **Fe reduction** | 0.5 ±0.0 – bottom | throughout | 0.5±0.0 – bottom | 0.5±0.0 – bottom | 0.8±0.6 – bottom |
| **Sulfate reduction** | surface – 5.8±2.0 | surface – 6.2±3.3 | surface – 11.7±3.1 | surface – 10.3±1.2 | surface – 11.0±2.0 |
| **Methanogenesis** | surface – bottom | surface – bottom | surface – bottom | 2.6±2.4 – bottom | 3.0±1.7 – bottom |