# Peer review of "Effects of eutrophication on sedimentary organic carbon cycling in five temperate lakes"

_Biogeosciences, 2019_

## Referee Comment (RC1) · Anonymous Referee #1 · 18 Apr 2019

General comments:

This ms deals with the impact of eutrophication on sediment TOC burial and mineralization in 5 temperate lakes. I find the topic interesting and the study and analyses are thoroughly performed and described.

However, I found that the introduction lacks some background information. First, the potential reasons for the increase of TOC burial in case of eutrophication appear too late and too briefly in the ms (p18 L10-15) and should be detailed in the introduction. The authors mostly assume that TOC accumulation/burial increased because of an increase of OC deposited on the sediment (e.g. p10 L25, P8 L1-2, p21 L8). Other potential reasons could be mentioned. For example, in some cases OC could be better preserved because of anoxic conditions due to low sediment mixing and low bioturba-

tion (so not necessarily related to an increase in NPP)? Secondly, besides the influence of mitigation strategies on OC burial (p3 L20), the novelty of the study regarding the relationships between eutrophication, TOC burial in general and respiration could be more detailed in the introduction. The authors mention respiration in the hypotheses p3 L28 but cite very few references on this aspect before that.

I think the authors should better justify the use of only one decay constant k for their calculations of OC burial and accumulation rates. The decomposition rate of TOC can strongly vary and the authors show in this ms that the respiration rates and ratios (RR DIC/ RR CH4) differ between sites.

Detailed comments:

P3 L30: I find strange to already give the results here

P4 L21: repetitive with L7-8

P4 L32: "The porewater was then sampled under strictly anoxic conditions" how? Because the syringe was rinced?

P7 L21: Punctuation missing

P8 L1-2: it can also mean that the decay constant is in reality slower (higher)?

P9 please provide more details on the flux (unit, the flux is from what to what...) L14 what is the unit of RR DIC and RR CH4? Please provide more details on how RR DIC and RR CH4 are calculated (i.e. write explicitly the relation between the flux, RR DIC and RR CH4).

---

## Referee Comment (RC2) · Anonymous Referee #2 · 4 Jun 2019

General comments

The paper studies sediment records of five different lakes that differ in trophic state and investigate the relationships between TOC accumulation, burial and historical P levels (eutrophication). I like the approach of comparing measured, modeled and reconstructed TOC to understand how sediment record was affected by eutrophication history. The MS is also informative by providing a nearly complete set of porewater chemistry data. However, I suggest the authors better describe the modeling methods, particularly the terms used (TOC accumulation, burial, TOC modeled, TOC reconstructed, etc.; see specific comments), which is important for the manuscript but poorly presented in the current version. I also find the results and discussions in cell counts, cell-specific rates less useful and largely speculative. I suggest reducing this part of

the discussion so that the paper has a better focus.

Specific comments: 1. Page 1, line 1: It's not obvious what is not well known. The introduction states that eutrophication increase TOC burial (page 3, line 9-10 and the references cited).

2. Page 1, line 25: I am not sure what "zonation of microbial respiration" means.

3. Page 1 line 29-30 "Instead, artificial lake ventilation, which is used to prevent water column anoxia in eutrophic lakes, may help sustain high rates of TOC burial and accumulation in sediments despite strongly reduced water column P concentrations." – This is speculative.

4. Page 30 – 35: very general statement. What insights?

5. Page 4, line 9- 21: I found the detail description here not necessary because Figure 1 is very self-explanatory. However, this is just a minor suggestion.

6. Page 5, line 11-12: Was air bubbling only done in oxygenated cores or all cores (e.g., cores under anoxia water)?

7. Page 7-8, Modeling of OC burial and accumulation rates through time: The calculation (equations) should be spelled out. It's not clear how what are calculated and how they are calculated. There seem to be multiple calculations here:

1) TOC modeled: using the surface TOC% to calculate the subsurface TOC% (based on the Middleburg power law), and comparing to measured subsurface TOC%. The purpose of this calculation is explained "...subsurface TOC% values that are higher ...", but it's better to spell out the equations, etc.

2) TOC reconstructed (but is this the same as TOC accumulation rate?) : using the TOC burial rate measured at depth to calculate TOC accumulation rate in the pass when the sediment of the specific depth was deposited at surface. The purpose of this calculation is not well explained. Also, the author should consider explaining the

calculations right after introducing TOC buried and TOC accumulation rate (line 9-11, equations 4 and 5).

These calculations and purpose of the calculations are important for the paper, but overall poorly described.

8. Equation 8: "n is porosity" should be mentioned here.

9. Page 10, line 15- 17: Is this based on TOC% in deeper sediments in Lakes Greifen and Zug? It's not clear.

10. Page 12, line 4-6: how were rates of aerobic respiration estimated? O2 fluxes?

11. Page 12, line 15-18: Bottom water NO3 may indicate denitrification rates in the water column, not necessarily denitrification in the sediments. Bottom water NO3 concentrations affects sediment NO3 penetrations, thus NO3 penetration depth in sediments is not a good indicator for sediment denitrification either. The authors may consider calculating NO3 fluxes, which is integrated rates (mmol/m2/d) for comparison.

12. Page 13, SO4: it may be interesting to compare SO4 fluxes at the SWI.

13. Page 15, line 30-31 "Thus, sediments deposited during the pre-eutrophication, early and mid eutrophication periods have similar RRtotal today...": it's not clear how this conclusion is reached?

14. Page 16, line 20-22 "Yet, even though water column P concentrations in most lakes have decreased close to pre- eutrophication levels since the ∼1970s, TOC burial and accumulation rates in eutrophic lakes remain significantly higher than before the eutrophication era (Figs. 3 and 4).": This is an interesting observation, and also agree with studies that show persisting high primary productivity after P reduction.

15. Page 16, line 23-25: "Despite the increase in TOC burial, lake sediments are not a static sink for OC. Increases in TOC accumulation and burial increase remineralization by stimulating microbial respiration (Fig. 6).": this is an interesting statement, however,

quantitative estimation is needed. With increasing TOC deposition (sedimentation), the net burial of TOC (long-term burial) may still be higher compared to pre-eutrophication period, even though microbial respiration increase.

16. Page 18, line 25-26 "Despite the observed decreases in TOC accumulations in two of the three eutrophic lakes, our calculated TOC accumulation rates for the period after peak eutrophication have remained well above those during pre-eutrophication times (Fig. 3).": Does primary productivity also decrease to pre-eutrophication times?

17. Page 19 Zonation and rates of dominant respiration processes: Rates were not quantified, and comparing zones in different sediments are less meaningful. I don't quite understand what the authors mean by "zonation". I suggest removing this part of the discussion.
* * *

---

## Author Comment (AC1) · 18 Jun 2019

General comments: This ms deals with the impact of eutrophication on sediment TOC burial and mineralization in 5 temperate lakes. I find the topic interesting and the study and analyses are thoroughly performed and described.

Answer: The authors thank the anonymous referee for the positive assessment and constructive suggestions. Below are our answers to the detailed comments of the referee. All changes to the ms in response to the referee comments are highlighted in

blue in the main document (please see link to the supplement below).

However, I found that the introduction lacks some background information. First, the potential reasons for the increase of TOC burial in case of eutrophication appear too late and too briefly in the ms (p18 L10-15) and should be detailed in the introduction.

Answer: We would like to refer the reviewer to the paragraph on page 3, lines 9-17, in which we explain the link between TOC burial and eutrophication.

The authors mostly assume that TOC accumulation/burial increased because of an increase of OC deposited on the sediment (e.g. p10 L25, P8 L1-2, p21 L8). Other potential reasons could be mentioned. For example, in some cases OC could be better preserved because of anoxic conditions due to low sediment mixing and low bioturbation (so not necessarily related to an increase in NPP)?

Answer: Thank you for this comment. We discuss oxygen exposure as a possibility on p. 18, L. 24 to p. 19, L. 2, as well as p. 19, L. 15-31. In this context we also discuss the apparent lack of a strong influence of bioturbation on C burial in eutrophic Lake Baldegg. Our analyses indicate that, although oxygen exposure may play a role, other variables appear to be more important drivers of OC burial.

Secondly, besides the influence of mitigation strategies on OC burial (p3 L20), the novelty of the study regarding the relationships between eutrophication, TOC burial in general and respiration could be more detailed in the introduction. The authors mention respiration in the hypotheses p3 L28 but cite very few references on this aspect before that.

Answer: We have changed the section introducing microbial respiration in the beginning of the Introduction (P. 2, L. 21-26) and added two more references. We have also rewritten the end of the introduction (p. 3 L. 24 – p. 4, L. 4) to more clearly highlight the novelty of the study.

I think the authors should better justify the use of only one decay constant k for their

calculations of OC burial and accumulation rates. The decomposition rate of TOC can strongly vary and the authors show in this ms that the respiration rates and ratios (RR DIC/ RR CH4) differ between sites.

Answer: We have tried to explain this more clearly in the Material and Methods (p. 8, L. 9 – p. 9, L. 4). Please note that – in the power model - the decay constant k changes as a function of time (it is not simply one constant). Furthermore, despite using the same k for TOC of comparable ages from different stations or lakes, the TOC decomposition rates differ greatly as a function of TOC content. We use the power model, which has been empirically shown to describe the relationships between TOC profiles and time across many sedimentary environments, as a reference, in order to identify the effects of eutrophication on TOC and TOC burial. We could have used different models to describe the TOC-time relationship, but inevitably these would have also failed to describe the observed subsurface peaks in TOC, as – same as in the power model - the common assumption is constant TOC deposition and burial. This assumption is wrong for lake sediments that have experienced eutrophication, as shown by this study and a small number of previous studies. With respect to the respiration rates, we would like to point out that we observe higher respiration rates in sediments of eutrophic lakes, where TOC is higher, compared to meso-/oligotrophic lakes (Figure 7). This is consistent with inferences regarding TOC that are based on the power model.

Detailed comments: P3 L30: I find strange to already give the results here

Answer: we use this to attune readers to the scientific trajectory of the following manuscript. This is a common practice.

P4 L21: repetitive with L7-8 Answer: Thank you, the repetition in L21 was removed as suggested.

P4 L32: "The porewater was then sampled under strictly anoxic conditions" how? Because the syringe was rinced?

Answer: a half sentence was added for clarification (p. 5, L. 4).

P7 L21: Punctuation missing

Answer: Punctuation was added.

P8 L1-2: it can also mean that the decay constant is in reality slower (higher)?

Answer: see answer to previous comment on the use of the decay constant.

P9 please provide more details on the flux (unit, the flux is from what to what...) L14 what is the unit of RR DIC and RR CH4? Please provide more details on how RR DIC and RR CH4 are calculated (i.e. write explicitly the relation between the flux, RR DIC and RR CH4).

Answer: a formula was added (Eq. 12) describing the calculations we made to calculate respiration rates from flux including units (P9 L16-18 and P10 L1-4). We furthermore added units to all equations.

Please also note the supplement to this comment:
https://www.biogeosciences-discuss.net/bg-2019-108/bg-2019-108-AC1-supplement.pdf

**Supplement:**

[revised manuscript text omitted]

15   by the decay curve (TOC$_{modeled}$) to investigate past differences in TOC burial relative to today. Accordingly, subsurface TOC$_{measured}$ values that are higher (lower) than TOC$_{modeled}$ indicate higher (lower) TOC burial rates than predicted by the power model. We calculate TOC burial rates according to

$$\text{TOC burial rate (g C m}^{-2}\text{yr}^{-1}) = \text{TOC}_{measured} \text{ (wt \%)} * \text{MAR (g m}^{-2}\text{yr}^{-1}) \qquad (4)$$

Higher (lower) subsurface TOC$_{measured}$ compared to T$_{modeled}$ are likely due to higher (lower) TOC accumulation rates at the time of deposition than today, and/or higher (lower) TOC burial efficiency than predicted by the power model. Assuming

20   that changes in TOC accumulation rate, e.g. due to changes in water column primary productivity over time, are the main driver of differences between TOC$_{measured}$ and TOC$_{modeled}$, we – for each sampled sediment horizon - reconstruct past TOC values (wt. %), for when this horizon was located at the sediment surface (TOC$_{reconstructed}$)

$$TOC_{reconstructed}(wt.\%) = TOC_{measured} (wt\%) + \left(\frac{(TOC\ loss\ (\%)*TOC_{measured}\ (wt\%))}{100}\right) \qquad \textbf{(5)}$$

where TOC loss (%) is the time-dependent percent fraction of TOC lost to mineralization according to the power model. We next reconstruct historic TOC accumulation rates, 
[revised manuscript text omitted]

---

## Author Comment (AC2) · 18 Jun 2019

General comments

The paper studies sediment records of five different lakes that differ in trophic state and investigate the relationships between TOC accumulation, burial and historical P levels (eutrophication). I like the approach of comparing measured, modeled and reconstructed TOC to understand how sediment record was affected by eutrophication history. The MS is also informative by providing a nearly complete set of porewater chemistry data.

Answer: The authors thank the anonymous referee for the positive review and the constructive suggestions. Below are our answers to the detailed comments. All changes to the ms in response to the referee comments are highlighted in blue in the main document (please see supplement of this document for the revised manuscript).

However, I suggest the authors better describe the modeling methods, particularly the terms used (TOC accumulation, burial, TOC modeled, TOC reconstructed, etc.; see specific comments), which is important for the manuscript but poorly presented in the current version.

Answer: Thank you for this comment. We have completely revised the text in the Materials & Methods in which we define these terms and hope it is more clear now (p. 8, l. 9 – p. 9, l. 4).

I also find the results and discussions in cell counts, cell-specific rates less useful and largely speculative. I suggest reducing this part of the discussion so that the paper has a better focus.

Answer: We have shortened the above sections. While perhaps some focus could be gained by removing them, as stated in the ms we wanted to not only look at burial of OC but also at mineralization processes in order to give a holistic picture. Not many studies have attempted to do this so far, especially not for lake sediments. Our study demonstrates that calculations and interpretations based on independently obtained data (TOC vs. porewater dissolved) and models are internally consistent and provide complementary insights. Furthermore, we would like to add that this manuscript is the foundation for several other manuscripts that will be published in the near future, to which we allude in the Discussion. The proposed interpretations are for the most part not speculative, but in line with cited previous studies, and/or supported by mentioned

unpublished data from our group. In the few cases that we speculate (e.g., p. 19, l. 24 & l. 23), we explicitly state this.

Specific comments 1. Page 1, line 1: It's not obvious what is not well known. The introduction states that eutrophication increase TOC burial (page 3, line 9-10 and the references cited).

Answer: We kindly refer this reviewer to the rewritten final paragraph of the Introduction. We hope it is much more clear now that this study is highly novel and lacks any similar precedent.

2. Page 1, line 25: I am not sure what "zonation of microbial respiration" means.

Answer: We have rewritten this sentence. For further background on the topic of zonation of microbial respiration reactions (aka "redox zonation") we refer to p. 2, l. 21-26, along with the listed references.

3. Page 1 line 29-30 "Instead, artificial lake ventilation, which is used to prevent water column anoxia in eutrophic lakes, may help sustain high rates of TOC burial and accumulation in sediments despite strongly reduced water column P concentrations." – This is speculative.

Answer: It is clear from our data that – among the three eutrophic lakes – Lake Baldegg, which has experienced the most stringent reductions in P concentrations and has been artificially mixed and aerated for ∼35 years to eliminate seasonal anoxia is the only one that has not experienced a significant decrease in TOC burial or accumulation since the period of peak eutrophication (Figure 3; p. 11, l. 27 to p. 12, l. 9; p. 18, l. 26 to p. 19, l. 31). We think it is important to mention this fact both in the Abstract and in the Discussion. If this trend holds true for other eutrophic lakes, then it is something that is important to know for management purposes.

4. Page 30 – 35: very general statement. What insights?

Answer: We assume that this reviewer was referring to the end of the Abstract (p. 1,

l. 30-35). We have replaced this statement with a more informative general summary statement.

5. Page 4, line 9- 21: I found the detail description here not necessary because Figure 1 is very self-explanatory. However, this is just a minor suggestion.

Answer: We respectfully disagree. We think this short accompanying text is necessary to highlight the most important differences in eutrophication histories between lakes.

6. Page 5, line 11-12: Was air bubbling only done in oxygenated cores or all cores (e.g., cores under anoxia water)?

Answer: Gentle air bubbling was done in all cores to prevent development of anoxic conditions and a stagnant water phase, even for the deep station of Lake Zurich, which presumably had hypoxic bottom water. This is a common practice that does not significantly affect the O2 profiles in sediments, as O2 measurements were done immediately after retrieval and were typically completed within one hour of sampling. We saw no major effect on sediment O2 profiles based on the fact that three successively measured O2 profiles at three different locations within each core were nearly identical, and did not show a time-dependent trend, e.g. an increase in O2 concentrations or penetration depth over time.

7. Page 7-8, Modeling of OC burial and accumulation rates through time: The calculation (equations) should be spelled out. It's not clear how what are calculated and how they are calculated. There seem to be multiple calculations here: 1) TOC modeled: using the surface TOC% to calculate the subsurface TOC% (based on the Middleburg power law), and comparing to measured subsurface TOC%. The purpose of this calculation is explained ". . .subsurface TOC% values that are higher . . .", but it's better to spell out the equations, etc. 2) TOC reconstructed (but is this the same as TOC accumulation rate?) : using the TOC burial rate measured at depth to calculate TOC accumulation rate in the pass when the sediment of the specific depth was deposited at surface. The purpose of this calculation is not well explained. Also, the author should

consider explaining the calculations right after introducing TOC buried and TOC accumulation rate (line 9-11, equations 4 and 5). These calculations and purpose of the calculations are important for the paper, but overall poorly described.

Answer: Thank you for this comment. We have completely revised this section of the text according to this comment (p. 8, l. 9 to p. 9, l. 7).

8. Equation 8: "n is porosity" should be mentioned here.

Answer: The authors agree and added this below the equation (P9 L14).

9. Page 10, line 15- 17: Is this based on TOC% in deeper sediments in Lakes Greifen and Zug? It's not clear.

Answer: Yes. This refers to the lowermost centimeters of the cores from Lakes Greifen and Zug (p. 11, l. 4-7), which were deposited during periods prior to the onset of eutrophication in these two lakes, have low TOC contents and are dominated by fine calcium carbonate-rich clay ("Seekreide"). This authigenic carbonaceous clay dominates sediments that were deposited prior to the period of eutrophication.

10. Page 12, line 4-6: how were rates of aerobic respiration estimated? O2 fluxes?

Answer: They were estimated based on O2 concentration gradients and O2 penetration depths. The O2 concentration gradients were generally steepest and the O2 penetration depths shallowest in eutrophic lakes. Given that these are diffusion-dominated sediments, this indicates that aerobic respiration rates were highest in eutrophic lakes. We have changed the structure of Figure 5 to make the concentration gradients of O2 (and nitrate) more clearly visible.

11. Page 12, line 15-18: Bottom water NO3 may indicate denitrification rates in the water column, not necessarily denitrification in the sediments. Bottom water NO3 concentrations affects sediment NO3 penetrations, thus NO3 penetration depth in sediments is not a good indicator for sediment denitrification either. The authors may consider calculating NO3 fluxes, which is integrated rates (mmol/m2/d) for comparison.

Answer: Yes, there certainly appears to be denitrification in the bottom water overlying the deep stations of Lake Greifen and Lake Zurich, as is stated in the text (p. 13, l. 4-7). However, there clearly also is denitrification in the sediments of all stations, or else nitrate would not be consumed within the top centimeters of sediment (see revised Figure 5). Hereby – as for O2 – the concentration gradients and penetration depths of nitrate allow us to make qualitative distinctions between the denitrification rates in sediments of different lakes. In order to make more quantitative predictions about denitrification rates (that take into account nitrification rates; nitrification appears to produce nitrate in surface sediments of several stations), it would have been essential to measure porewater nitrate concentrations at higher depth resolution. We plan to do this in the future.

12. Page 13, SO4: it may be interesting to compare SO4 fluxes at the SWI.

Answer: We agree that this would be interesting to do in one of our next studies, but it is beyond the focus of this manuscript.

13. Page 15, line 30-31 "Thus, sediments deposited during the pre-eutrophication, early and mid eutrophication periods have similar RRtotal today...": it's not clear how this conclusion is reached?

Answer: We think this is very clear based on this figure, but suspect that perhaps the text is not clear. Thus, we have added in parentheses "i.e. from 1840-1960" to explicitly which sediments have similar RRtotal, despite having very different original TOC accumulation rates.

14. Page 16, line 20-22 "Yet, even though water column P concentrations in most lakes have decreased close to pre- eutrophication levels since the _1970s, TOC burial and accumulation rates in eutrophic lakes remain significantly higher than before the eutrophication era (Figs. 3 and 4).": This is an interesting observation, and also agree with studies that show persisting high primary productivity after P reduction.

Answer: We agree.

15. Page 16, line 23-25: "Despite the increase in TOC burial, lake sediments are not a static sink for OC. Increases in TOC accumulation and burial increase remineralization by stimulating microbial respiration (Fig. 6).": this is an interesting statement, however, quantitative estimation is needed. With increasing TOC deposition (sedimentation), the net burial of TOC (long-term burial) may still be higher compared to pre-eutrophication period, even though microbial respiration increase.

Answer: Our measurements (Figure 5) and quantitative estimations (Figures 6 and 7) show clearly that mineralization rates are higher in eutrophic lakes. Nonetheless, as stated throughout the manuscript, TOC accumulation and burial rates are also highest in the eutrophic lakes (Figures 2-4). Therefore, net mineralization rates are not high enough to override differences in the original amounts of TOC that were deposited and subsequently buried.

16. Page 18, line 25-26 "Despite the observed decreases in TOC accumulations in two of the three eutrophic lakes, our calculated TOC accumulation rates for the period after peak eutrophication have remained well above those during pre-eutrophication times (Fig. 3).": Does primary productivity also decrease to pre-eutrophication times?

Answer: Publicly available data from BAFU data indicate a decrease but continuously high phytoplankton concentrations in the eutrophic lakes. Presumably this elevated primary productivity is a prerequisite for the continuously high TOC accumulation rates. We discuss possible drivers behind the sustained high TOC accumulation, e.g. P remobilization from sediments as a driver of continued high primary productivity, in the Discussion (P19, L. 3-31).

17. Page 19 Zonation and rates of dominant respiration processes: Rates were not quantified, and comparing zones in different sediments are less meaningful. I don't quite understand what the authors mean by "zonation". I suggest removing this part of the discussion.

Answer: We refer this reviewer to the Introduction section (p. 2, l. 21-26) where we introduce the concept of redox zonation and provide extensive literature on redox zonation and the underlying concepts. We especially recommend the provided references by Froelich et al. 1979, Jørgensen & Kasten 2006, Canfield et al. 2005, and Canfield & Thamdrup 2009.

The purpose of our analyses was not to quantify rates of different redox reactions, but to investigate their distributions with respect to each other. It is a central (yet contested) dogma in the field of sediment biogeochemistry that microbial respiration reactions are separated into zones based on the energetically most favourable available electron acceptor. Most past studies investigating this dogma have focused on marine sediments. We wanted to determine if it applied to lacustrine sediments, and if trophic state, due to its effects on electron donor availability (TOC), has an impact on redox zonation. Our results indicate that there is no clear separation of dominant respiration reactions into "redox zones" in any of the lakes. This is an important finding for our understanding of what controls (or does not control) the distribution of OC terminal mineralization reactions.

Please also note the supplement to this comment:
https://www.biogeosciences-discuss.net/bg-2019-108/bg-2019-108-AC2-supplement.pdf

**Supplement:**

[revised manuscript text omitted]

15   by the decay curve (TOC$_{modeled}$) to investigate past differences in TOC burial relative to today. Accordingly, subsurface TOC$_{measured}$ values that are higher (lower) than TOC$_{modeled}$ indicate higher (lower) TOC burial rates than predicted by the power model. We calculate TOC burial rates according to

$$\text{TOC burial rate (g C m}^{-2}\text{yr}^{-1}) = \text{TOC}_{measured} \text{ (wt \%)} * \text{MAR (g m}^{-2}\text{yr}^{-1}) \qquad (4)$$

Higher (lower) subsurface TOC$_{measured}$ compared to T$_{modeled}$ are likely due to higher (lower) TOC accumulation rates at the time of deposition than today, and/or higher (lower) TOC burial efficiency than predicted by the power model. Assuming

20   that changes in TOC accumulation rate, e.g. due to changes in water column primary productivity over time, are the main driver of differences between TOC$_{measured}$ and TOC$_{modeled}$, we – for each sampled sediment horizon - reconstruct past TOC values (wt. %), for when this horizon was located at the sediment surface (TOC$_{reconstructed}$)

$$TOC_{reconstructed}(wt.\%) = TOC_{measured} (wt\%) + \left(\frac{(TOC\ loss\ (\%)*TOC_{measured}\ (wt\%))}{100}\right) \qquad \textbf{(5)}$$

where TOC loss (%) is the time-dependent percent fraction of TOC lost to mineralization according to the power model. We next reconstruct historic TOC accumulation rates, 
[revised manuscript text omitted]

---

## Author Response (AR2)

**"Effects of eutrophication on sedimentary organic carbon cycling in five temperate lakes" by Annika Fiskal et al.**

**Point by point response to reviews**

**Anonymous Referee #2**

Most of my comments for the previous version were addressed. The authors have improved the Method section substantially, particularly the model calculations and definition of terms. This makes the paper easier to understand. More explanation on comparison of nitrate reduction rates is now included. My major concern is the conclusion about "redox zonation". However, I consider this an easy fix by clarifying the terms, rephrasing the conclusions and some parts of discussion (see specific comments below)

The authors thank the anonymous referee for the positive and helpful comments. We have addressed all open concerns as shown below and in the markup version of the ms.

1. Page 1, line 25: "No major effect of trophic state on the relative distributions of these reactions" – I disagree (see comment # 5).

We agree that this sentence was not written in a clear manner and changed it now see Page 1, line 25 to 26

2. Page 1, line 30-32: This is interesting the explanations on reduced P concentration is not clear (see comment # 6)

We agree and tried to clarify this sentence in the ms (page 1, line 30-32)

3. Page 3, line 14-15: What's the difference between "OC loading" and "OC sedimentation rates"? What exactly is OC loading? From external?

We agree that this might be confusing and tried to clarify this term in the text (page 3 line 14-15).
We refer to OC loading as the input of OC, mainly through water column primary production. OC sedimentation rates are the amounts of OC that arrive at the sediment surface over time. Not all OC produced in surface water by photosynthesis makes it to the sediment surface, since a significant fraction is mineralized in surface water or during sinking.

4. Page 3, line 22-23: "Furthermore, links between historically documented human activities in lakes and lake watersheds, eutrophication in lake water columns, which would inform on links between OC burial and mineralization processes in lake sediments." – There is a grammatical problem (incomplete sentence). Also, it's not obvious why "these links" would inform "those links".

We have deleted this sentence, since its content is already covered in the following paragraph.

5. Page 17: line 7-10: "The eutrophication-related changes in respiration rates are, however, not reflected in the vertical zonation of microbial respiration pathways with respect to each other " – I still don't understand what the authors mean by

10    this. Do you mean the zones are not clearly separated? Please state that specifically.

"the depth intervals of denitrification, Mn reduction, Fe reduction…. Show strong overlaps, independent of lake trophic state" – It's not clear why there should be a link between the trophic state and "zonation". What do you expect the effects of trophic state would be? Would you expect they not overlapping when TOC sedimentation is high? Why? More clarification is needed. Otherwise, I suggest rephrasing the sentence, perhaps using "regardless of their trophic state". "Independent" suggests not

15    having any effect, but I would argue that eutrophication would shift the locations of each zone. However, by looking at porewater profiles now, one would not know if the dynamics in primary production (and TOC sedimentation) shift the reaction zones. This is because dissolved species move (diffuse) much faster than sediment burial. The porewater profiles you are seeing now represent a snapshot of what's going on right now, although the solid sediments is a mixture of historical and fresh sediments.

Thank you for this valuable comment. We have modified the text to include an additional explanatory sentence (page 17, line 2-5) and have replaced "independent" with "regardless".

6. Page 19, line 10-14: "Such remobilization of P is not necessarily evident based on total P concentrations as most remobilized

25    P is present in the highly bioavailable inorganic form" – but total P includes bioavailable inorganic P (the P released from sediments) and P from wastewater. Changes in sediment released P and its products (e.g., after being taken up as biomass P) would be reflected in the dynamics of total P.

Thank you for this comment. We have removed these sentences, since they are speculative and not necessary. There is ample

30    evidence from other lakes, that P remobilization sustains high rates of primary production in lakes for long time periods after drastic reductions in P inputs have been made (we cite numerous studies that have documented this).

7. Page 21, line 31 "Trophic state does not affect the zonation of microbial respiration reactions" – this conclusion is misleading (see comments above). I would rephrase it to: We observe less clear zonation of microbial redox reactions in sediments of lakes, regardless of their trophic states. I suggest not using the word "independent".

5  We have changed the sentence as suggested and added some further explanations to improve the clarity.

**List of relevant changes**

- We rewrote the two sentences of the abstract (p.1, L. 25-26 and p.1, L. 30-33) to make our statement about redoxzonations more clear as well as to better describe the relationship of P concentration, artificial ventilation and carbon burial.

- We changed one sentence of the introduction in order to clarify better what we mean with OC loading and OC sedimentation (p. 3, L. 13-15).

- Additionally we deleted one sentence, that was not clear mentioned by the referee (p.3, L.22-23) as this topic is covered more clear in the following paragraph.

- We modified and added a sentence to the paragraph in the discussion section about the overlaps of microbial respiration reactions, as suggested by the referee (p. 17, L 1-6).

- We also modified this part in the conclusion section (p. 21, L 21-25) to make the section on microbial respiration reactions more clear and replaced the word independent with regardless as suggested by the referee.

[revised manuscript text omitted]